# Source differences in the components and cytotoxicity of PM$_{2.5}$ from automobile exhaust, coal combustion, and biomass burning contributing to urban aerosol toxicity

**Xiao-San Luo[1,#,\*], Weijie Huang[1,#], Guofeng Shen[2], Yuting Pang[1], Mingwei Tang[1], Weijun Li[3], Zhen Zhao[1], Hanhan Li[1], Yaqian Wei[1], Longjiao Xie[4], Tariq Mehmood[5]**

[1]International Center for Ecology, Meteorology, and Environment, School of Applied Meteorology, Nanjing University of Information Science & Technology, Nanjing 210044, China
[2]Laboratory of Earth Surface Processes, College of Urban and Environmental Sciences, Peking University, Beijing 100871, China
[3]Department of Atmospheric Sciences, School of Earth Sciences, Zhejiang University, Hangzhou 310027, China
[4]Health Science Center, Peking University, Beijing 100871, China
[5]College of Ecology and Environment, Hainan University, Haikou 570228, China

*Correspondence*: Xiao-San Luo (xsluo@nuist.edu.cn)

[#]Authors contributed equally to this work

**Abstract.** Although air quality guidelines generally use the atmospheric concentration of fine particulate matter (PM$_{2.5}$) as the metric for air pollution evaluation and management, the fact can't be ignored that different particle toxicity is unequal and significantly related to their sources and chemical compositions. Therefore, judging the most harmful source and identifying the toxic component would be helpful to optimize air quality standards and prioritize targeted PM$_{2.5}$ control strategies to protect public health more effectively. Since the combustions of fuels, including oil, coal, and biomass, are main anthropogenic sources of environmental PM$_{2.5}$, their discrepant contributions to health risks of mixed ambient aerosol pollution dominated by respective emission intensity and unequal toxicity of chemical components need to be identified. In order to quantify the differences among these combustion primary emissions, ten types of PM$_{2.5}$ from each typical source group, i.e., vehicle exhaust, coal combustion, and plant biomass (domestic biofuel) burning, were collected for comparative study with toxicological mechanisms. Totally thirty type individual combustion samples were inter-compared with representative urban ambient air PM$_{2.5}$ samples, which chemical characteristics and biological effects were investigated by component analysis (carbon, metals, soluble ions) and *in vitro* toxicity assays (cell viability, oxidative stress, inflammatory responses) of human lung adenocarcinoma epithelial cells (A549). Carbonaceous fractions were plenteous in automobile exhaust and biomass burning, while heavy metals were more plentiful in PM$_{2.5}$ from coal combustion and automobile exhaust. The overall ranking of mass-normalized cytotoxicity for source-specific PM$_{2.5}$ was automobile exhaust > coal combustion > domestic plant biomass burning > ambient urban air, possibly with differential toxicity triggers, that the carbonaceous fractions (organic carbon, OC; elemental carbon, EC) and redox-active transition metals (V, Ni, Cr) assisted by water-soluble ions (Ca$^{2+}$, Mg$^{2+}$, F$^{-}$, Cl$^{-}$) might

play important roles in inducing cellular reactive organic species (ROS) production, causing oxidative stress and inflammation,
resulting in cell injury and apoptosis, thus damage human health. Coupled with the source apportionment results of typical
urban ambient air $PM_{2.5}$ in eastern China, reducing toxic $PM_{2.5}$ form these anthropogenic combustions will be greatly beneficial
to public health. Besides the air pollution control measures that have been implemented, like strengthening the vehicle emission
standards, energy switching from coal to gas and electricity, and controlling the open incineration of agricultural straws, further
methods could be considered especially through preferentially reducing the diesel exhaust, then lessening the coal combustion
by replacement with low-ash clean coals, and depressing the rural crop straw biomass burning emissions.

## 1 Introduction

As a mixture of multiple sources, ambient particulate matter (PM) arise from anthropogenic activities are continuously
deteriorating the urban air quality, particularly in developing countries. Among these, fine PM with an aerodynamic diameter
of less than 2.5 μm ($PM_{2.5}$) is recognized as a serious public health concern due to its long persistence in air, carcinogenicity
and acute toxicity to humans (Al-Kindi et al., 2020). There were extensive epidemiological evidences that airborne PM can
cause serious negative effects on human health, such as respiratory and cardiovascular diseases, genetic mutations, and
developmental disorders (Chowdhury et al., 2022;Clemens et al., 2017;Lelieveld et al., 2021;Smith, 2021). Currently, either
the world air quality guidelines or the national air quality standards use the mass concentration of $PM_{2.5}$ as the metric for $PM_{2.5}$
pollution evaluation and management, however, the particle toxicity are unequal and significantly related to their sources and
chemical compositions varying with space and time (Shiraiwa et al., 2017). Therefore, to identify which component(s) and
source(s) of ambient PM are most harmful to health, will be helpful to evaluate air quality and prioritize targeted PM control
strategies for protecting public health more effectively.
Besides natural sources, most aerosols come from anthropogenic activities especially energy consumption, including the
combustion of fossil fuels causing industrial emissions and automobile exhaust, and biomass burning (McDuffie et al.,
2021;Wu et al., 2022). These diverse sources make the ambient air $PM_{2.5}$ become a complex mixture with multiple chemical
components, such as salts, organic carbon (OC), elemental carbon (EC), mineral and trace metals (Bari and Kindzierski, 2016).
The physiological mechanisms of PM-induced cell toxicity in respiratory system have been continuously investigated with
some progresses (Li et al., 2022b;Kelly and Fussell, 2012, 2020;Shiraiwa et al., 2017;Mack et al., 2019), such as the metabolic
activation, oxidative stress, inflammatory response, and apoptosis, focused on by current study. In brief, after inhalation and
deposition onto the epithelium, redox-active materials in $PM_{2.5}$ can induce the release of reactive organic species (ROS), which
cause oxidative stress (an imbalance between ROS and antioxidants, i.e., disequilibrium of the redox state of a cell) followed
by inflammation and cell death. The ROS can mediate subsequent signaling pathways leading to biomolecule damage (e.g.,
DNA, lipid, and protein) and cellular injury, through mediating inflammatory responses including the release of pro-
inflammatory cytokines like IL-6 and TNF-α by epithelial cells (Ahmed et al., 2020;Landwehr et al., 2021). For instance,
oxidative stress could trigger the induction of pro-inflammatory transcription factors, such as nuclear factor (NF)-κB, via the
mitogen-activated protein kinase (MAPK) signaling pathway. Components adsorbed on particle surface, such as redox-active
metals (transition metals, Fe, Ni, V, Cr, Cu), organic compounds (polycyclic aromatic hydrocarbons, PAHs; quinones), or
even carbonaceous core of particles, are responsible for oxidative stress (Ahmed et al., 2020;Cachon et al., 2014). The non-
redox active metals (Zn, Pb, Al) can also influence the toxic effects of transition metals by exacerbating or lessening the
production of free radicals. The EC may not be a directly toxic component of $PM_{2.5}$ but rather operate as a universal carrier of
combustion-derived chemicals (semi-volatile organic fractions, transition metals) of varying toxicity (Kelly and Fussell, 2020).
Inorganic soluble sulphates and nitrates are acidic and can interact with and influence the solubility other compositions like
metal bioavailability (Fang et al., 2017;Weber et al., 2016). However, besides the well-known toxic pollutants in environment
like heavy metals and PAHs, which specific components and which particular sources are the most critical factors dominating
the ambient aerosols' health risks, still need be explored.
Past studies performed in various countries have focused on physicochemical characterization or biological effects of
ambient air $PM_{2.5}$ respectively (Weagle et al., 2018;Jia et al., 2017;Wang et al., 2020). For example, the source analysis of
$PM_{2.5}$ by photochemical modelling (Bao et al., 2018), chemical composition of regional $PM_{2.5}$ (Chi et al., 2022), and the
mechanism of $PM_{2.5}$ toxicity was independently reported recently (Jia et al., 2020). Because differences in particle composition,
sources, and toxicity appear in different urban environments (Zhao et al., 2019;Borlaza et al., 2018), the source profiles of
different emission inventories were needed to elucidate the local aerosol pollution characteristics for control strategies. For
instance, it was reported that increased hospital admission risks were significantly associated with sources of vehicle exhaust,
coal combustion, and secondary inorganic aerosols; in particular, coal combustion was positively correlated with increases in
mortality risks (Du et al., 2021). Coal combustion and vehicle exhaust contributed more significantly to cancer risks of
respiratory exposure to atmospheric heavy metals in Tianjin city of the northern China during the cold seasons (31% and 11%)
than the warm seasons (11% and 4%) (Tian et al., 2021); while in Nanjing city of the eastern China, traffic emissions and non-
traffic combustion (coal/waste/biomass) contributed 35% and 31% to carcinogenic risks of urban $PM_{2.5}$-associated metals
respectively (Xie et al., 2020). Traffic was suggested playing the most crucial role in enhancing the toxicity of fine particles
(Park et al., 2018). The particle composition of motor vehicle exhaust was related to automobile types with various fuels,
engines, and loads (Lin et al., 2020). A strong catalytic reactivity of metals in PM emitted from diesel vehicles was observed
by dithiothreitol (DTT) assay (Jesus et al., 2018). It was found that straw burning during the harvest season is a major trigger
of severe air pollution in many regions (Sahu et al., 2021). Aerosols from open biomass burning in the Amazon had a stronger
ability to induce ROS than laboratory-generated secondary organic aerosols (Tuet et al., 2019). Although there were emerging
studies on particle emission from single source, quantitatively comparative studies on multi-source pollutants as well as the
differential composition and unequal toxicity of various sources are still limited.
The main objective of current study was to compare the chemical components and corresponding mass-normalized
toxicological effects of individual $PM_{2.5}$ from various combustion sources and their unequal contributions to ambient aerosol
health risks. The aim is to provide experimental evidences supporting the targeting control of specific anthropogenic sources
with prominent risks based on their pivotal toxic components. Therefore, we collected both representative ambient PM$_{2.5}$
samples (n = 16) from urban air and typical source PM$_{2.5}$ samples (n = 30) from automobile exhaust, coal combustion, and
plant biomass burning. Their independent profiles of chemical compositions and *in vitro* cytotoxicity (cell viability, oxidative
stress, and inflammatory responses) were investigated and intercompared, to assess the differences in source-to-receptor
toxicity and to infer the core toxic components and respective harmful contribution. The pivotal toxic components were
identified based on the source-sink bi-directional composition-effect results, which were further used to assess the health
toxicity contribution of various emission sources to ambient air PM$_{2.5}$, supported by its source apportionment through positive
matrix factorization (PMF) and chemical mass balance (CMB) models.
**2 Materials and methods**
**2.1 Collection of PM$_{2.5}$ samples from primary emissions of 30 typical combustion sources and from representative**
**ambient urban air**
Totally 30 types of primary PM$_{2.5}$ samples emitted directly from automobile exhaust, coal combustion, and plant biomass
(domestic biofuel) burning were respectively collected as follows for both chemical and toxicological analyses.
A total of 10 types of vehicles were chosen for exhaust investigation. They were further categorized into 7 sub-groups,
including small duty gasoline coaches (SDGCs), small duty diesel coaches (SDDCs), middle duty diesel coaches (MDDCs),
heavy duty diesel coaches (HDDCs), light duty diesel vans (LDDVs), middle duty diesel vans (MDDVs), and heavy duty
diesel vans (HDDVs). The detailed information of these representative local automobiles was showed in Table S1.
To cover all coal types consumed in the city, 10 representative types of coal were gathered for investigation. They were
further classified into 4 sub-groups, including 2 types of honeycomb coal (HC), 3 types of anthracite coal (AC), and 2 types
of bituminous coal (BC) mainly for restaurant or household use, and 3 types of industrial coal (IC) for coal-fired power plants
and steel-smelting industry. The detailed characteristic to physical-chemical of these typical coals purchased from local market
were showed in Table S2.
Considering the plant biomass combustion in rural areas surrounding the megacity, 10 representative types of agricultural
and forestry solid wastes were gathered for investigation. Straws of rice, wheat, corn, soybean, peanut, rape, and sesame,
corncob, branches of peach and pine, were selected as plant biomass fuels and further divided into 2 sub-groups, including 8
types of crop straw and 2 types of firewood. The detailed characteristic analysis of these typical plant biomass fuels collected
from rural areas around Nanjing city were showed in Table S3.
The PM$_{2.5}$ samples directly emitted from these combustion sources were collected by dilution channel sampling method
(Figure S1), using a 4-channel particulate matter dilution sampler (HY-805, Hengyuan Technology Development Co., CN).
Each sampling included 3 parallel channels of quartz microfiber filter (Figure S2) and 1 channel of Teflon membrane filter
with diameters of 47 mm, through a size selector for PM$_{2.5}$ with a flow rate of 160 L min$^{-1}$ (each channel is 40 L/min). Clean
air was pumped for 10 min before and after each sample was collected. Before using, the blank quartz filters were incinerated
by a muffle furnace at 500 ℃ for 3 h to remove any possible organic matters, while Teflon filters were baked at 60 ℃ for 4 h.
After being equilibrated in a constant temperature and humidity chamber for 24 h, the filters were weighed both before and
after sampling for gravimetric measurements, then the mass of collected PM$_{2.5}$ could be calculated. The sampled filters were
stored in a refrigerator at -20 ℃ before analysis. The quartz filter loaded PM$_{2.5}$ samples were used for carbon and ion analysis,
and for toxicity tests, while the parallel Teflon filter loaded samples were used to determine metals.
As the actual mixture of various source particles in real environment, totally 16 ambient air PM$_{2.5}$ samples (each time lasting
23h) covering a year monthly were collected from December 2019 to October 2020 in an urban site surrounded by traffic,
residential and commercial quarters of Nanjing city, Yangtze River Delta of eastern China, using a high-volume air sampler
(800 L min$^{-1}$) with quartz microfiber filters (Li et al., 2022a).

## 2.2 Chemical composition analysis

All collected source and ambient PM$_{2.5}$ samples were conducted following component analysis (Li et al., 2023). For the
concentrations of heavy metals in particulates, samples were digested by concentrated HNO$_3$-HClO$_4$ acids with a progressive
heating program and determined by inductively coupled plasma optical emission spectrometry (ICP-OES; Optima8000,
PerkinElmer, for Cr, Mn, Ni and Pb), with elements (V, Co, As) at lower concentrations measured by ICP mass spectrometry
(ICP-MS; NexIONTM300X, PerkinElmer). Blank filter, reagent blank, replicates, and standard reference material (NIST SRM
1648a, urban dust) were adopted for analytical quality control, with recoveries ranged 90-110 %. Carbonaceous species (OC
and EC) in PM$_{2.5}$ were determined using a DRI-2001A OC/EC (Atmoslytic Inc., Calabasas, CA, USA). For the concentrations
of water-soluble ions (WSIs), the main cations (Na$^+$, K$^+$, Mg$^{2+}$, Ca$^{2+}$, NH$_4^+$) and anions (NO$_3^-$, SO$_4^{2-}$, Cl$^-$, F$^-$) in PM$_{2.5}$ were
measured by ion chromatography (IC, Thermo Fisher Scientific, USA), using the Metrosep C6-150/4.0 column for cations and
the Metrosep A Supp 5 150/4.0 column for anions, respectively.

## 2.3 Preparing mass-normalized PM$_{2.5}$ suspension for cell exposure

Totally 30 source and 16 ambient PM$_{2.5}$ samples were also performed cytotoxicity tests. In order to elute the particles
completely from the quartz membranes, a whole PM$_{2.5}$-loaded sample filter was cut into small pieces, immerged in ultrapure
water and extracted six times (30 min for each) in an ultrasonic bath at 0 ℃. Although the ultrasonication might impact the
ROS (Miljevic et al., 2014), the inevitable systematical error was ignored in this study. The extract was then suction filtered
through a 2.6 μm pore-size nylon membrane to remove possible quartz fragments, and the bulk filtrate was freeze-dried back
to pure PM$_{2.5}$ powder. Ultimately, based on particle mass, the gathered PM$_{2.5}$ was dispersed by sterile phosphate-buffered
saline (PBS) to a concentration of 400 mg L$^{-1}$, and then diluted to PM$_{2.5}$ suspension of 80 mg L$^{-1}$ with serum-free Dulbecco's
modified eagle medium (DMEM) for following *in vitro* cell exposure (Li et al., 2022a).

## 2.4 Cell culture and cellular toxicity tests by *in vitro* PM$_{2.5}$ exposure

Aerosol pollution can harm lung alveoli and epithelial cells, and the A549 adenocarcinoma epithelial cell has long been used as a suitable epithelial alveolar model (Li et al., 2022b;Park et al., 2018). The A549 cells were cultured in RMPI-1640 medium (Gibco, USA) supplemented with 10% fetal bovine serum (FBS, Hyclone, USA) and 1% antibiotic penicillin-streptomycin (100 U mL$^{-1}$) at 37 ℃ in a 5% CO$_2$ incubator. After PM$_{2.5}$ exposure, cell viability and the indicators reflecting oxidative damage and inflammatory responses were determined respectively. While the cell viability assay was helpful in determining PM$_{2.5}$ dose to cells, the endogenous ROS measurements revealed the status of cellular oxidative potential after PM$_{2.5}$ exposure followed by the relative effects of ROS on various stages of cellular toxicity like inflammatory responses (Gali et al., 2019). The cell viability (metabolic activity) was evaluated by mitochondrial activity and determined by the methyl-thiazol-tetrazolium (MTT) assay (Chen et al., 2019). After trypsin action, the density of cells in the logarithmic growth phase was adjusted to $1 \times 10^5$ mL$^{-1}$. Cell suspensions were inoculated into 96-well plates (Costar, USA) at 100 μL per well. The blank control well (without medium and PM$_{2.5}$ suspension) and reagent control well (with medium but without PM$_{2.5}$ suspension) were set together. After incubation for 24 h and removing the cellular supernatant, various types of PM$_{2.5}$ suspension (concentration of 80 mg L$^{-1}$) were added to 96-well plates and incubated for 24 h. Based on pre-experiments, the oxidative stress and inflammation response sensitively under this dose, while the cell viability can keep sufficient. Fresh medium and MTT reagent (Solarbio, Beijing, CN) were added to each well and the supernatant was discarded, then 100 μL of formazan lysate was added to each well. The optical density (OD) values were measured at 490 nm using a microplate reader (Thermo MULTISKAN FC, USA). Cell viability (%) = (OD$_{treatment}$ − OD$_{blank\ control}$) / (OD$_{reagent\ control}$ − OD$_{blank\ control}$). The levels of cellular ROS production causing oxidative stress in cells, pro-inflammatory cytokines including tumor necrosis factor-alpha (TNF-α) and interleukin-6 (IL-6) production for determining the expression of genes related to the inflammatory response in the supernatant were analyzed by enzyme-linked immunosorbent assay (ELISA) kits (Jiangsu Meibiao Biotechnology Co., Ltd., CN), and OD values were measured at 450 nm (Huang et al., 2020;Pang et al., 2020).

## 2.5 Data analysis

The statistical analysis was performed by IBM SPSS statistics 24 and plotted by Origin 2020b software. Spearman correlation coefficients were produced by the correlation analysis. The variance was statistically significant when the statistical test level was $p < 0.05$, and extremely significant when $p < 0.01$. Statistical analyses were performed using Kruskal-Wallis test (Kruskal and Wallis, 1952).

The source apportionment of PM$_{2.5}$ mass in urban ambient air was conducted by the receptor models PMF (EPA PMF version 5.0) and CMB (EPA CMB 8.0). All measured constituents (OC, EC, Cu, Cr, Co, Ni, As, Pb, Mn, V, Na$^+$, K$^+$, Mg$^{2+}$, Ca$^{2+}$, NH$_4^+$, Cl$^-$, F$^-$, NO$_3^-$, and SO$_4^{2-}$) were selected as PMF model input data, and a four-factor solution was chosen as the optimal solution based on an assessment of the interpretability of the source profiles and the seasonal variability of the source contributions. Due to the high concentration of sulfate and nitrate in ambient PM$_{2.5}$, and being lack of specific actual source to

emit sulfate and nitrate, we added the virtual source profiles of secondary sources in CMB model (Table S4). The virtual source
profiles of secondary sources are represented by the proportion of sulfate, nitrate and ammonium in pure ammonium sulfate
and ammonium nitrate.
**3 Results**
**3.1 Contributions of combustion primary sources to urban ambient air PM$_{2.5}$**
As shown in Figure S3, although have been significantly improved with the national air quality in recent years, the estimated
annual PM$_{2.5}$ concentrations of representative city Nanjing (59.1 $\pm$ 20.5 $\mu$g m$^{-3}$) was 1.7 times higher than the China national
standard (35 $\mu$g m$^{-3}$) and 11.8 times higher than the WHO guidelines (5 $\mu$g m$^{-3}$). Urban air PM$_{2.5}$ pollution levels in the cold
season were higher than the warm season. The similar source apportionment results from PMF and CMB models are illustrated
in Figure 1. Four major sources of the ambient PM$_{2.5}$ were produced by the PMF model (Figure S4), including secondary
aerosols, and primary particles of automobile exhaust, coal combustion, and plant biomass burning, which account for 34.0%,
27.7%, 25.2%, and 13.1% of total PM$_{2.5}$ mass concentration, respectively. The CMB model source profiles are shown in the
Table S4, and we normalized contribution of secondary aerosols (32.4%), and automobile exhaust (32.2%), coal combustion
(25.1%), plant biomass burning (10.3%). Therefore, although the contribution of secondary aerosols cannot be ignored, the
main anthropogenic sources of urban air PM$_{2.5}$ were primary emissions from the various fuel combustions.
**3.2 Chemical compositions of different PM$_{2.5}$ from 30 combustion sources and from representative urban ambient air**
Typical chemical components including carbonaceous fractions, heavy metals, and WSIs of all PM$_{2.5}$ samples from both
ambient air and combustion sources were analyzed and compared with each other.
According to the comparisons of PM$_{2.5}$ bound carbonaceous fractions (Figure 2), automobile and biomass sourced PM$_{2.5}$
contained significantly higher total carbon (TC) content than coal combustion and ambient air, while the OC/EC ratio trend
was ambient air > coal combustion > biomass burning > automobile exhaust sources. It indicated that the carbon content of
ambient PM$_{2.5}$ mixture was lower and dominated by OC than that of combustion primary sources, implying that the OC in
ambient air may be aged or cleaned. The OC undergoes various chemical reactions in the atmosphere, such as oxidization by
ozone and hydroxyl radicals, resulting in degradation. Figures S4-S7 showed the detailed carbon fraction characteristics
(contents and ratio) of PM$_{2.5}$ from each specific source. Carbonaceous fractions in automobile exhaust PM$_{2.5}$ were high but the
difference between OC and EC content was small. Depending on the diverse automobile fuels, loads and tailpipe emission
standards, the concentrations of carbon fractions in exhaust PM$_{2.5}$ varied widely with vehicle categories. The carbonaceous
portion of PM$_{2.5}$ gradually declines as emission regulations rise, and EC likewise declines dramatically (Figure S5). However,
such differences among coal types were less, except the bituminous coal with extreme high OC (Figure S6). The carbonaceous
fraction of PM$_{2.5}$ from domestic plant biomass burning differed in raw material species that tree branches source PM$_{2.5}$
generally contained higher carbon contents than those from crop straws (Figure S7).
Based on the grouped (Figure 3) and individual (Figures S9-S12) distributions of the measured heavy metals in various
PM$_{2.5}$, the V concentrations of combustion sources were generally higher while Co and Mn were lower than ambient urban air.
Coal combustion emissions carried highest levels of Pb and were enriched in Cu and As (Figure S10), while biomass burning
were rich in Cr and Ni (Figure S11). However, automobile exhausts were enriched in most heavy metals, especially Cu, and
Cr, Ni, V, Mn (Figure S9). Heavy metals from different types of automobile exhausts with the same emission standard varies
greatly. Anthracite and industrial coal combustions contain similar heavy metals much more than bituminous coal. Generally,
Pb, V, Mn, As, and Cu in branches source PM$_{2.5}$ were higher than straws, while Cr, Ni, and Co were dominant and higher in
straw burning emissions. A special discovery was that corn cob burning PM$_{2.5}$ carried more heavy metals than corn straw and
was the biomass with the highest emission levels of heavy metals. Correspondingly, ambient air PM$_{2.5}$ were also rich in most
metals, especially Mn, Pb, and Ni, Cu, Cr. Therefore, coal combustion sources might contribute most Pb to urban ambient air,
and contribute significant Cu and As with automobile exhaust emissions, while plant biomass burning and automobile sources
contribute the Cr and Ni. Besides natural dust, automobile exhaust should be the main anthropogenic source of airborne Mn.
Considering the PMF source apportionments of ambient aerosols, automobile exhaust should be the main source of Cr in urban
air PM$_{2.5}$, and also the source for Cu together with coal combustion.
According to the comparisons of water-soluble cation and anion concentrations in various PM$_{2.5}$ (Figure 4), coal
combustions contained highest $SO_4^{2-}$ and $NH_4^+$, automobile exhausts had highest contents of $NO_3^-$, $Na^+$ and $Ca^{2+}$, while plant
biomass burning sources contained highest $K^+$ and $Cl^-$, but $Mg^{2+}$ was the lowest for all sources. However, the urban ambient
air PM$_{2.5}$ contained highest $NO_3^-$ and were also dominated by $SO_4^{2-}$ and $NH_4^+$, for which $NO_3^-$ should be mainly contributed
by secondary aerosols and automobile primary source, $SO_4^{2-}$ and $NH_4^+$ should be significantly from coal combustions. Besides
$NO_3^-$, $Na^+$ and $Ca^{2+}$, automobile source PM$_{2.5}$ also had the highest $F^-$ and $Mg^{2+}$ concentrations than other sources. The detailed
concentration distributions of WSIs in PM$_{2.5}$ from each specific source were provided in Figures S12-S14. The WSIs levels
vary widely with specific source categories. PM$_{2.5}$ from LDDVs-2 had the lowest amount of WSIs compared to the other
automobile exhausts (Figure S13). Similar to the metal composition, bituminous coal also had the lowest WSIs among all coals
(Figure S14). Compared to branches, PM$_{2.5}$ from burning crop straws had much greater levels of $K^+$, $Cl^-$, $SO_4^{2-}$ and less levels
of $F^-$, $NO_3^-$ (Figure S15).
To summarize, the overall concentrations of measured TC, cumulated heavy metals and WSIs in PM$_{2.5}$ from each source
type were showed in Figure 5. Among all source emission and environmental receptor samples, the cumulated heavy metals
from coal combustion was highest and automobile exhaust was higher than ambient PM$_{2.5}$, the overall carbon contents from
automobile exhaust and biomass burning were both higher than ambient PM$_{2.5}$, while only the cumulated soluble ions in PM$_{2.5}$
from primary source of coal combustion was equivalent to the ambient aerosols. In a word, chemical compositions of PM$_{2.5}$
distributed much diversely and varied significantly with the specific source types of combustion emissions.

**3.3 Cell viability, oxidative stress and inflammation levels exposed to various mass-normalized PM$_{2.5}$**

Multiple toxicological endpoints (cell viability, oxidative stress, and inflammation) that facilitate identifying the specific particle triggering ROS and inflammatory responses resulting in cell death were evaluated for source-specific PM$_{2.5}$. After 24 h exposure to the same dose of different PM$_{2.5}$ obtained from specific emission sources, the A549 lung cells also showed varied toxicological responses (Figure 6). The survival rate of cells exposed to automobile exhaust PM$_{2.5}$ was much lower than ambient air PM$_{2.5}$ by 16.6% (Figure 6a). Automobile exhaust PM$_{2.5}$ induced the highest ROS production in cells higher than biomass burning, which was 26.4% and 14.8% higher than ambient PM$_{2.5}$ (Figure 6b). Coal combustion induced the highest cellular IL-6 production followed by automobile exhaust, which was 13.1% and 4.48% higher than ambient air PM$_{2.5}$, respectively; while the PM$_{2.5}$ from automobile exhaust and biomass burning induced similarly 10.4% higher cellular production of TNF-α than ambient PM$_{2.5}$ (Figure 6c, 6d). These results suggested that, combustion primary emission PM$_{2.5}$ had stronger ability to induce oxidative stress and inflammatory injury in lung cells than ambient air PM$_{2.5}$, thus resulted in the higher probability of apoptosis induction (Victor and Gottlieb, 2002;Wang et al., 2013). Generally, the mass-normalized PM$_{2.5}$ from primary source of automobile exhaust posed the strongest overall toxicity. Therefore, to protect public health by controlling PM$_{2.5}$ pollution, these anthropogenic combustions were key target sources, especially the most toxic automobile PM$_{2.5}$ should be reduced preferentially.

**3.4 Correlations between various PM$_{2.5}$ components and toxicity endpoints**

Spearman correlation coefficients between chemical compositions and cellular toxicological response indicators were applied to screen the key components of all PM$_{2.5}$ involved in cell injury (Figure 7). It was found that, the degrees of correlations varied with the toxicological mechanisms of different airborne chemicals. Based on the overall PM$_{2.5}$ samples from various sources, the pro-inflammatory cytokine IL-6 showed significantly strong positive correlations with some heavy metals (As, Pb, V, Cu), while TNF-α and oxidative stress (ROS) had similar significantly positive correlations with aerosol components of carbon fractions (EC, OC) and transition metals (V, Cr, Ni). The TNF-α also showed positive correlation with water soluble Cl$^-$ and K$^+$, and ROS correlated with F$^-$, Ca$^{2+}$ and Mg$^{2+}$.

**4 Discussion**

**4.1 Chemical markers for source apportionments of ambient air PM$_{2.5}$**

Combustion emissions are key anthropogenic sources contributing to urban air PM$_{2.5}$, through both primary and secondary aerosols, which were 66% and 34% estimated by PMF model, 67.6% and 32.4% by CMB model, respectively (Figure 1). Compared to the PMF results, the proportions of coal combustion and secondary sources in the CMB results show minimal changes, while biomass contributions are slightly underestimated, and there is a slight increase in the proportion attributed to vehicular emissions. The high concentrations of chemical markers are usually used in source analysis, such as ammonium

sulfate and nitrate for secondary aerosols which are originated mainly from the gaseous precursors (e.g., $NH_3$, $SO_2$ and $NO_X$)
(Mahilang et al., 2021), the EC, Cu, Mn, and Ni for vehicle exhaust (Srivastava et al., 2021) , the As, Pb, OC, EC, $SO_4^{2-}$ and
relatively low $NO_3^-/SO_4^{2-}$ ratios for coal combustion (Dai et al., 2020), soluble $K^+$ and $Cl^-$ for plant burning (Jain et al., 2020).
The detailed chemical species of these specific source emission $PM_{2.5}$ samples also supported the results. Moreover, low
OC/EC ratio of high TC content, high $NO_3^-$, $F^-$, $Na^+$, $Ca^{2+}$ and $Mg^{2+}$, V and Mn of automobile exhaust; Pb and As, $SO_4^{2-}$ and
$NH_4^+$ of coal combustion; soluble $K^+$ and $Cl^-$, and high OC/EC ratio of high TC for plant biomass burning found in current
study (Figures 2-5), could also be corresponding potential aerosol source markers.

### 4.2 Common $PM_{2.5}$ components related to specific combustion sources

Generally, the automobile exhaust $PM_{2.5}$ had high TC content and low OC/EC value with considerable EC content (Figure 2),
varying with specific vehicle types (Figure S5-8). The contents of the carbon fractions from diesel vehicles were 2.39 times
more than gasoline exhausts (Figure S5), and the OC/EC ratios of diesel exhausts were 37.3% of gasoline vehicles, owing to
both considerable contents of EC and OC from diesel vehicle emission $PM_{2.5}$. Some diesel vehicles showed higher EC
emissions with age, so exhaust cleaning devices for them are suggested. In addition, the amounts of OC and EC in exhausts
gradually decreased with the strengthened emission standards they met (Wong et al., 2020). In $PM_{2.5}$ samples obtained from
coal combustion (Figure S6), the TC contents of bituminous coals was 3.97, 6.41, and 11.6 times higher than that of honeycomb
coals, anthracite coals, and industrial coals, respectively, because bituminous coals contain higher volatile fraction. Emissions
of non-methane VOCs increase with the volatile content of the coal (He et al., 2022). The vast majority of organic aerosols
from bituminous coal are generated in the ignition and fierce combustion phases, which account for 99.9% of the entire
combustion process; while these two phases of anthracite coal generate only 77% of the entire process (Zhou et al., 2016).
Moreover, as the volatile matter in the coal decreases, the temperature at which weight loss begins and ends shift to higher
values, that may be due to the lower amount of aliphatic chains present. It has been reported that for bituminous maximum
weight loss happens in the range 490–600 ℃, while in the case of anthracites coals it occurs between 750 and 870 ℃ (De la
Puente et al., 1998). Therefore, besides the way of combustion and the use of combustion stoves, the coal quality related to
different coal types and origins determine the carbonaceous fractions of the PM emitted by coal combustion (Zhang et al.,
2022). In the $PM_{2.5}$ samples from plant biomass combustion (Figure S7), OC contents were 2.21 times higher than EC contents,
except that pine branches contained higher EC and rapeseed straw had considerable contents of EC and OC. The OC in ambient
$PM_{2.5}$ dominated the carbonaceous component (Figure S8), consistent with the North China Plain and Indo-Ganges Plain
(Flores et al., 2020;Xu et al., 2019). Combining the TC contents and OC/EC ratios, carbonaceous components in ambient $PM_{2.5}$
mainly originate from semi-volatile organic compounds (SVOCs) (Wang et al., 2018). Previous studies have reported that
carbonaceous aerosols are mainly originated from fossil fuel combustion in transportation, coal combustion in power plants
and industries, and biomass combustion (Kang et al., 2018;Zhang et al., 2015). Thus, to control ambient carbon aerosol
pollution, besides reducing the precursor emissions of secondary organic aerosols (SOA), controlling primary aerosols
especially EC from diesel vehicles might be effective measures.
Airborne redox-active metals are usually linked with the oxidation stress of $PM_{2.5}$. Different types of automobiles emitted
diverse metal contents (Figure S9). Metal elements in automobile exhaust are primarily contributed by fuels, lubricants, and
engine component abrasion. Because Mn is a common antidetonator that delays and prevents the oxidation of hydrocarbons
and increases the octane number, which not only increases the thermal efficiency of the engine but also improves the emission
performance of the vehicle (Cheung et al., 2010), the Mn content was greater in gasoline vehicle exhausts than in diesel
vehicles. Although there are multi-sources of traffic Pb emissions such as fuel combustion and brake wear (Wang et al.,
2019;Panko et al., 2019), the automobile exhaust Pb content of gasoline vehicles were greater than diesel vehicles owing to
oil combustion. Moreover, for the same vehicle type (LDDVs-1 and 2; HDDVs-1 and 2; SDGCs-1 and 2), the stricter the
emission standard required, the lower the exhaust metal contents. The metal contents in the $PM_{2.5}$ of trucks was higher than
that of passenger cars (Wu et al., 2016). In the combustion $PM_{2.5}$ of 10 coal types (Figure S10), Pb contents were the highest
than other heavy metals, similar to available findings (Zhang et al., 2020). The $PM_{2.5}$ metals from bituminous coal were
significantly lower than other coal types, because indicated by the coal quality analysis, bituminous coal has a low ash content
which is mainly derived from non-combustible minerals in coal. These findings suggested that coal maturity might be an
important factor influencing the metal composition of particulates emitted from coal combustion (Shen et al., 2021;Zhang et
al., 2021). Heavy metal contents in biomass burned $PM_{2.5}$ varied much widely with raw plant types (Figure S11), although
dominated by Cr and Ni. Different plant species and even different plant parts differ significantly in their ability to uptake and
accumulate metals from soil (Zhao et al., 2020). Moreover, because of the high enrichment factors of some metals for crop
straws (Zhang et al., 2016;Sun et al., 2019), they also released more Cr, Ni, and Co during burning than fuelwoods. Total metal
emissions were highest in corn cob but lowest in peanut straw burning $PM_{2.5}$. The heavy metals enriched in urban ambient air
$PM_{2.5}$ showed slightly seasonal pattern (Figure S12), while contents of V, Co, and As were relatively low and less affected by
seasonal changes. Accordingly, supported by the metal profiles of anthropogenic combustion sources and ambient aerosols, to
control the environmental airborne heavy metal pollution, key targets might be the Pb, Cu and As from honeycomb, anthracite
and industrial coal combustion, Cu from vehicle exhausts and especially V from light duty diesel van with the CN.III emission
standard and Mn from gasoline vehicles, Cr and Ni from biomass especially crop straws burning.
Epidemiological studies have also shown the mortality closely related to the WSIs such as sulfate and nitrate in aerosols
(Ostro et al., 2009;Liang et al., 2022). Among the WSIs contents of various automobile exhaust $PM_{2.5}$ (Figure S13), $NO_3^-$ and
$Ca^{2+}$ were the most abundant anion and cation, respectively. The high $NO_3^-$ in the automobile $PM_{2.5}$ may be due to $NO_x$
production during high-temperature combustion, while the high $Ca^{2+}$ content should be related to additives in automobile fuels
and calcium-based lubricants (Hao et al., 2019;Yang et al., 2019). Moreover, the exhaust WSIs decreased with the strengthened
automobile emission standards required. Coal combustion $PM_{2.5}$ contained relatively higher $SO_4^{2-}$ and $NH_4^+$ concentrations
followed by $Cl^-$ than other WSIs species (Figure S14). Among various coal types, industrial coals emitted highest $SO_4^{2-}$
followed by honeycomb and industrial coal with also high $NH_4^+$, but bituminous coals emitted low WSIs which were mainly
$NO_3^-$, $F^-$ and $Na^+$, $Ca^{2+}$. The WSIs emission factors of honeycomb coal were generally higher than those of lump coal (Yan et
al., 2020). For biomass combustion emissions (Figure S15), $Cl^-$ and $K^+$ were dominant WSIs in $PM_{2.5}$ from straw-type fuels
(Tao et al., 2016;Sillapapiromsuk et al., 2013), but fuelwood-type combustion emitted high $NO_3^-$. Plant species absolutely
determine the emissions (Liao et al., 2021). Finally, there were also high levels of $NO_3^-$, $SO_4^{2-}$, and $NH_4^+$ in ambient air $PM_{2.5}$
(Zhang et al., 2019) (Figure S16), even higher than the investigated combustion sources, so other sources like the secondary
aerosols may also contribute. Consequently, target combustion primary aerosols WSIs might include, the $NO_3^-$ from vehicle
exhausts and fuelwood burning; $SO_4^{2-}$ and $NH_4^+$ from honeycomb, anthracite and industrial coal combustion; $Cl^-$ and $K^+$ from
biomass especially crop straw burning.

### 355 4.3 $PM_{2.5}$ toxicity related to specific sources by pivotal chemical components

The complexity of the sources and compositions of atmospheric $PM_{2.5}$ leads to different toxicological effects (Newman et al.,
2020;Kelly, 2021). The toxicological effects of $PM_{2.5}$ are not comparable among different studies owing to distinct exposure
concentrations, biological models, endpoints, and $PM_{2.5}$ generation methods (Kelly and Fussell, 2020;Park et al., 2018). In this
study, we employed same exposure conditions and biological endpoints, in order to obtain comparable toxicity data for $PM_{2.5}$
from different sources. Our mass-normalized results demonstrated that automobile exhaust $PM_{2.5}$ induced the highest lethality
and cellular ROS and TNF-α production, coal combustion $PM_{2.5}$ induced the highest cellular IL-6 production, plant biomass
burning $PM_{2.5}$ induced considerable cellular TNF-α and ROS production (Figure 6). Generally, various toxicities of
combustion emission primary $PM_{2.5}$ were much greater than the urban ambient air $PM_{2.5}$ (Figure 6), owing to the higher
concentrations of specific toxic components in $PM_{2.5}$ from these sources. The supplementary information had included
exhaustive cytotoxicity indicators from each individual source (Figure S17-S20). While the survival rate of cell exposed to
CN.III emission standard $PM_{2.5}$ was the lowest and the capacity to induce cells to produce ROS was the highest for CN.IV,
automobile exhaust had a similar potential to cause cells to produce inflammatory cytokines (Figure S17). The capability to
induce IL-6 production in cells was highest for industrial coal $PM_{2.5}$, whereas bituminous coal had the highest survival rate of
cells and TNF-α induction capacity (Figure S18). From the Figure S19 we can see that the $PM_{2.5}$ cytotoxicity of straws and
branches burning was analogous, but it should be noted that the cell viability of various straw $PM_{2.5}$ differs significantly, that
may be related to the raw fuel characteristics.
These possible mechanisms were implied by the overall relationships between the measured chemical components with
cytotoxicity indicators of $PM_{2.5}$ from various specific sources (Figure 7). In general, both TNF-α and ROS were significantly
positively correlated with carbonaceous fractions and redox-active transition metals (V, Cr, Ni), which were main contributors
of automobile exhausts and biomass burning. The IL-6 was significantly positively correlated with some heavy metals (As and
Pb, V and Cu), which were main contributors of coal combustion sources. Potential mechanisms include that, carbon fractions
bound in $PM_{2.5}$ could be transformed into reactive metabolites and then induce ROS production in cells (Stevanovic et al.,
2019), and the PM$_{2.5}$ bound transition metals could also induce ROS production through the Fenton reaction and disrupt the
function of enzymes in cells (Verma et al., 2010;Sørensen et al., 2005). Oxidative stress can lead to inflammatory infiltration
of neutrophils and stimulate immune cells to produce inflammatory cytokines, among which TNF-α and IL-6 play important
roles in the inflammation development (Xu et al., 2020). Ultimately, excessive production of ROS leads to dysfunctional
endoplasmic reticulum responses and dysfunctional lipid metabolism in ROS bursts can result in cell membrane damage and
even cell death (Piao et al., 2018;Zhao et al., 2004). There have been some related supporting reports. For instance, the OC
and EC were significantly associated with biological responses of PM from vehicle emissions collected in tunnels (Niu et al.,
2020). The polar or quinone fractions of PAHs in diesel engine exhaust particles significantly contributed to the heightened
toxic response (Xia et al., 2004). The PM$_{2.5}$ generated from biomass burning contained a substantial concentration of
carbonaceous components. In addition, Cr and Ni in PM$_{10}$ from straws were highly associated with ROS (Li et al., 2023). In
current study, cellular ROS was also correlated with water soluble Ca$^{2+}$, F$^-$, and Mg$^{2+}$, which were main contributors of
automobile exhaust PM$_{2.5}$. The Ca$^{2+}$ controls the membrane potential and regulates mitochondrial adenosine triphosphate (ATP)
production, and excessive Ca$^{2+}$ leads to energy loss and more ROS production (Madreiter-Sokolowski et al., 2020). Moreover,
the TNF-α was also positively correlated with water soluble Cl$^-$ and K$^+$, which were main contributors of plant burning PM$_{2.5}$.
Therefore, the accumulations of some organic matters with high carbonaceous content (OC, EC) in PM$_{2.5}$ typically from
automobile exhausts and plant biomass burning, redox-active metals (V, Cr, Ni) and water-soluble anions (Cl$^-$, F$^-$) and cations
(Ca$^{2+}$, Mg$^{2+}$) contributed by various combustions, might induce ROS production in cells, cause cellular damage through
oxidative stress and inflammatory responses, impair cell viability and finally harm human health.
Considering the multi-endpoints measured and the PM$_{2.5}$ toxicity mechanisms mentioned above, based on the cell viability
first, and then ROS followed by inflammatory markers, together with the significantly related toxic chemical composition
contents (Park et al., 2018), we put forward a general sequence of overall mass-normalized toxicity for these combustion
source PM$_{2.5}$ to managers. To improve the urban environmental air quality for better public health benefits by controlling
aerosols pollution, considering the differential toxicity intensity of each chemical component and their contributions from
various sources to ambient aerosols, preferential targets of specific primary PM$_{2.5}$ sources and bound pollutants from
anthropogenic combustions are suggested as following sequence: reducing the automobile exhaust PM$_{2.5}$ containing high
contents of EC, transition metals (V, Cu, Ni, Cr), and ions (Ca$^{2+}$, Mg$^{2+}$, F$^-$, Na$^+$) from diesel exhausts by strengthening the
emission standards and accelerating the phasing out of highly polluting vehicles; then lessening the coal combustion rich in
heavy metals (As, Pb, Cu) by replacement with low-ash clean coals; and depressing the biomass burning containing high OC,
Ni, Cr, Cl$^-$ and K$^+$ from rural crop straw emissions and promoting domestic cleaner energy such as natural gas.
**4.4 Limitations and perspectives**
In current study, we selected A549 cell based on previous abundant experimental experiences and also because it has been
used popularly in *in vitro* toxicology studies to elucidate the cellular and molecular mechanisms of PM involved in lung for
many decades (Li et al., 2022b). However, recently the human normal bronchial epithelial cell BEAS-2B was preferred over
the human lung adenocarcinoma epithelial cell A549. For instance, both cells were used in an aerosol study (Bonetta et al.,
2017), results of which highlighted the higher sensitivity of BEAS-2B cells respect to A549 also in samples with low level of
pollutants, because the $PM_{0.5}$ samples from Italian towns can induce genotoxicity in normal cells while cancer cells might be
resistant to their adverse effects. Therefore, although our results are reasonable under the same exposure conditions, there were
still potential limitations of A549 cells since they may be more resistant to exposure to external compounds, and the generally
more sensitive BEAS-2B cells are suggested for future studies.
In toxicity assessments, cell vitality reflects the overall health of cells, encompassing factors such as cell membrane integrity,
intracellular metabolic activity, and cell proliferation capacity. Decreased cellular vitality may be associated with cell damage,
toxic effects, or cellular apoptosis. Inflammation markers are employed to assess the extent and nature of inflammatory
reactions, including the production of cytokines and inflammatory mediators, as well as the activation status of inflammatory
cells. Inflammation is a complex physiological response, typically delineated by the immune and inflammatory reactions of
the body to stimuli such as injury or infection. Alterations in inflammation markers can indicate the intensity and nature of the
inflammatory response. In this study, multiple biological responses of epithelial cells to various $PM_{2.5}$ were evaluated,
including that, cell viability evaluated the mitochondrial dehydrogenase activity of the living cells, excessive intracellular ROS
formation induced by $PM_{2.5}$ was responsible for oxidative stress to the cells, cytokines IL-6 and TNF-α were determined for
the effect of $PM_{2.5}$ on pro-inflammatory response in cells. In general, in vitro data can be used to rank various types of particles
in terms of the toxic potential including possible carcinogenicity. Each marker will help to understand the hazard and toxicity
of $PM_{2.5}$. However, the toxicity of $PM_{2.5}$ may be the result of multiple components acting through disparate physiological
mechanisms, with inconsistent relationships among endpoints (Park et al., 2018). For instance, in BEAS-2B cells, oxidative
stress generated by $H_2O_2$ exposure often results in cytotoxicity rather than by stimulating cytokine/chemokine responses,
sometimes no correlation between oxidative damage and cytokine/chemokine responses. Moreover, TNF-α gene was not
detected in BEAS-2B cells exposed to atmospheric PM collected from Benin, but the gene expression of other inflammatory
cytokines (IL-1β, IL-6, and IL-8) were significantly induced, and decreasing cell viability was highly correlated with high
secretion of all studied cytokines (Cachon et al., 2014). Therefore, in the present study, it was impossible to analyze all
chemicals in $PM_{2.5}$ and determine all related toxicological endpoints, so unmeasured chemicals and endpoints might also play
roles in the incongruous or unexplained results, and we also can't over-explain the mechanisms just based on statistical
relations. To overcome these hurdles, standardization of toxicological studies (experimental methodologies) and reporting
guidelines are necessary for tracking and comparing results.
This study ranked the unequal "toxic effects" based on the same mass concentration of $PM_{2.5}$ exposure in body lung fluid
system, while the "health risks" usually relating to the inhalation exposure concentration of $PM_{2.5}$ in ambient air were not
calculated and evaluated quantitatively. Moreover, non-linear concentration-response functions for various endpoints and
different exposure concentrations might also limit using toxicological data straightforwardly to predict health effects
(morbidity, mortality) in human populations, so drawing conclusions precisely quantifying/ranking the health risks of $PM_{2.5}$
from specific sources or of individual $PM_{2.5}$ components is still not an easy task (Kelly and Fussell, 2020). Therefore, coupled
with source apportionment and exposure level of ambient aerosols pollution, toxicology combined with epidemiology studies
linking these factors and indicating scientific mechanisms would help to reach conclusions.
Moreover, the exact effective measures to control these specific key toxic components from the emissions of various
combustion sources indeed a challenge, but still need to be explored. The findings of this research provide a specific direction
for better air pollution control and public health. Besides the environmental technological methods of controlling toxic
components targeting source materials, combustion processes, and final emissions, the environmental management policies
are also beneficial to such aims, like the choice of fuel types, especially for the management of domestic biomass fuel burning.
For examples, potential solutions include promoting new green energy vehicles and low-ash clean coals, depressing the diesel
exhaust and rural crop straw burning emissions.

## 5 Conclusions

In current study, we found that 2/3 mass of urban ambient air $PM_{2.5}$ in a typical megacity of eastern China originated from
primary sources of anthropogenic combustions including coal, automobile, and biomass. Because of the significant differences
in the chemical compositions, the diverse $PM_{2.5}$ from both mixed ambient air and directly from individual combustion sources
showed much differential mass-normalized *in vitro* toxicity to the human lung epithelial cells, either for the environmental
aerosol samples collected from different seasons, or for the primary emissions of $PM_{2.5}$ from various specific source types.
According to the comparative study and correlation analysis, the carbonaceous fractions (OC, EC) and redox-active heavy
metals (V, Ni, Cr) assisted by water-soluble ions ($Ca^{2+}$, $Mg^{2+}$, $F^-$, $Cl^-$) might play important roles in inducing cellular ROS
production, causing oxidative stress and inflammation, resulting in cell injury and apoptosis, thus damage human health. These
toxic pollutants accumulated in specific-source $PM_{2.5}$ varied by the emission types and raw fuel properties. Combined with
chemical composition and general cytotoxicity rank, the preferential controlling targets of specific combustion sources might
be automobile exhaust (diesel vehicles with emission standards inferior to CN.IV), coal combustion (high ash and high sulfur
coals), and rural plant biomass burning (crop straws). Although showing the synthetic effects of mixed compositions and
complex sources, besides preventing the secondary aerosols from combustions, preferentially targeted reductions of toxic
$PM_{2.5}$ direct emissions from these primary sources, would produce great benefits for public health with improved ambient air
quality. Overall, the chemical findings of our toxicological research could help to support the precise, oriented, effective,
efficient, and economical composition-source-based strategies for urban aerosols pollution control. However, as a prospect,
the detailed mechanisms for unequal toxicity of PM with complicated components from various sources and their quantitative
contributions to the health effects of ambient air $PM_{2.5}$ mixture still need in-depth study.

## Supplementary materials

There are 20 figures (Figure S1-S20) and 3 tables (Table S1-S4) in the Supporting Information.

## Data availability

All raw data can be provided by the corresponding authors upon request.

## Author contributions

XSL conceived and supervised the study; WH, YP, MT, HL, and ZZ collected the samples; WH, YP, MT, WL, HL, ZZ, GS, and LX analyzed the chemical compositions; WH, YP, and MT performed the toxicity tests; WH, YP, MT, and XSL analyzed the data; WH and XSL wrote the manuscript draft; XSL, WH, GS, and TM reviewed and edited the manuscript.

## Competing interests

The authors declare that they have no conflict of interest.

## Financial support

This work was supported by the National Natural Science Foundation of China (NSFC 41977349, 41471418).

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

# Captions of figures

**Figure 1.** Source contributions (%) to the urban ambient air $PM_{2.5}$ (models PMF vs CMB).

**Figure 2.** Carbon contents (mg $kg^{-1}$) and ratio in $PM_{2.5}$ from various specific sources (n=10 for each combustion source and n=16 for urban ambient air).

**Figure 3.** Heavy metal contents (mg $kg^{-1}$) in $PM_{2.5}$ from various specific sources (n=10 for each combustion source and n=16 for urban ambient air).

**Figure 4.** Water-soluble ion (WSI) contents (mg $kg^{-1}$) in $PM_{2.5}$ from various specific sources (n=10 for each combustion source and n=16 for urban ambient air).

**Figure 5.** Cumulated typical measured components (mg $kg^{-1}$) in $PM_{2.5}$ from various specific sources (n=10 for each combustion source and n=16 for urban ambient air).

**Figure 6.** Cell viability, oxidative stress and inflammation levels of human alveolar epithelial cell lines (A549) exposed to $PM_{2.5}$ suspension (80 mg $L^{-1}$) from various specific sources (n=10 for each combustion source and n=16 for urban ambient air).

**Figure 7.** Overall correlations between typical cellular toxicological responses and chemical compositions of $PM_{2.5}$ from various sources (*$p < 0.05$,#$p < 0.01$; n=46).

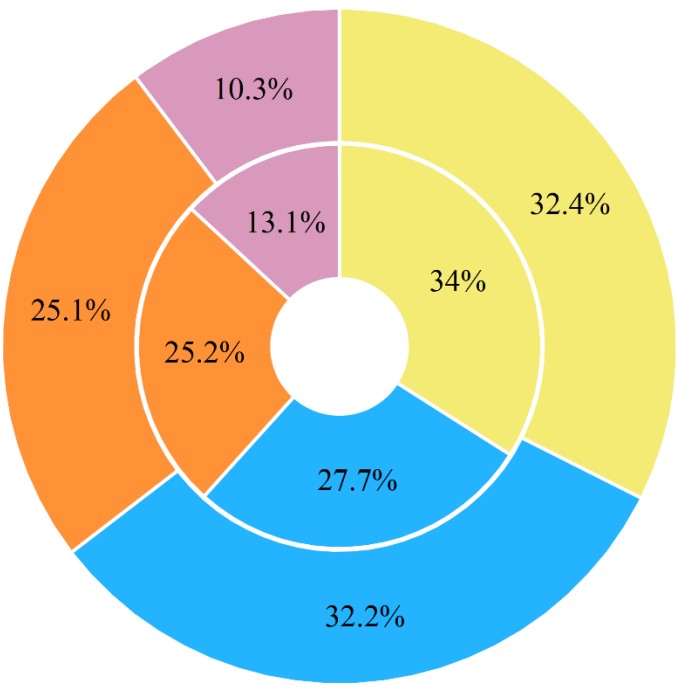

Inner ring - outer ring:PMF model -CMB model

**Figure 1: Source contributions (%) to the urban ambient air PM$_{2.5}$ (models PMF vs CMB).**


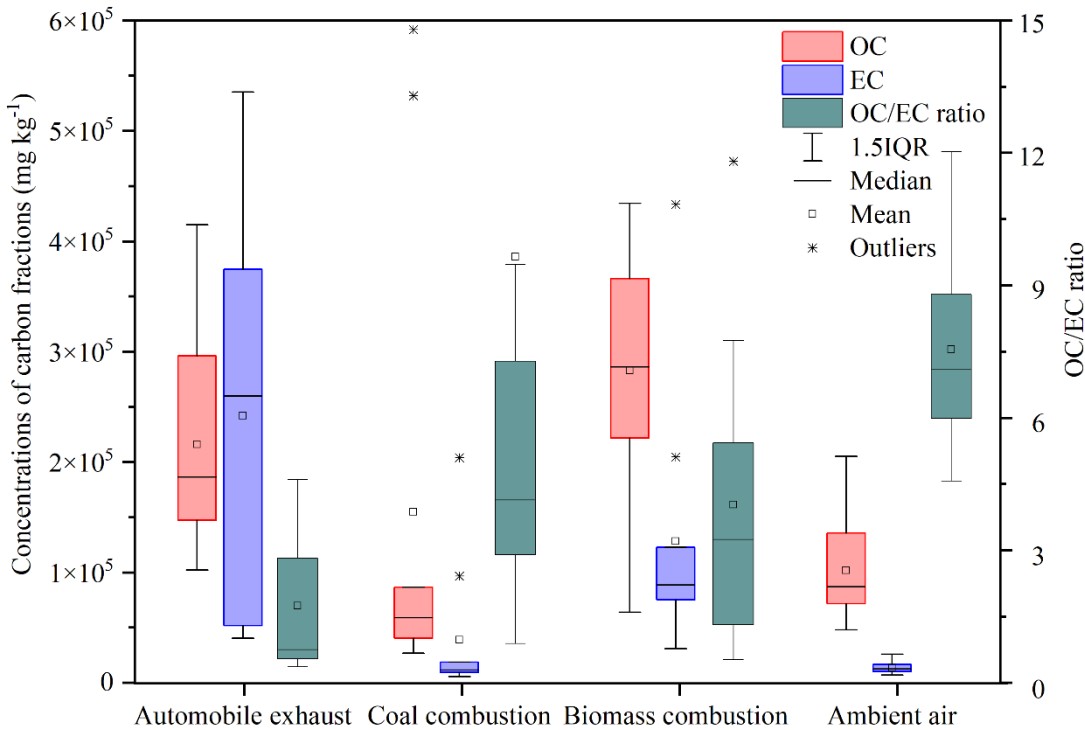

**Figure 2: Carbon contents (mg kg$^{-1}$) and ratio in PM$_{2.5}$ from various specific sources (n=10 for each combustion source and n=16**
**for urban ambient air).**

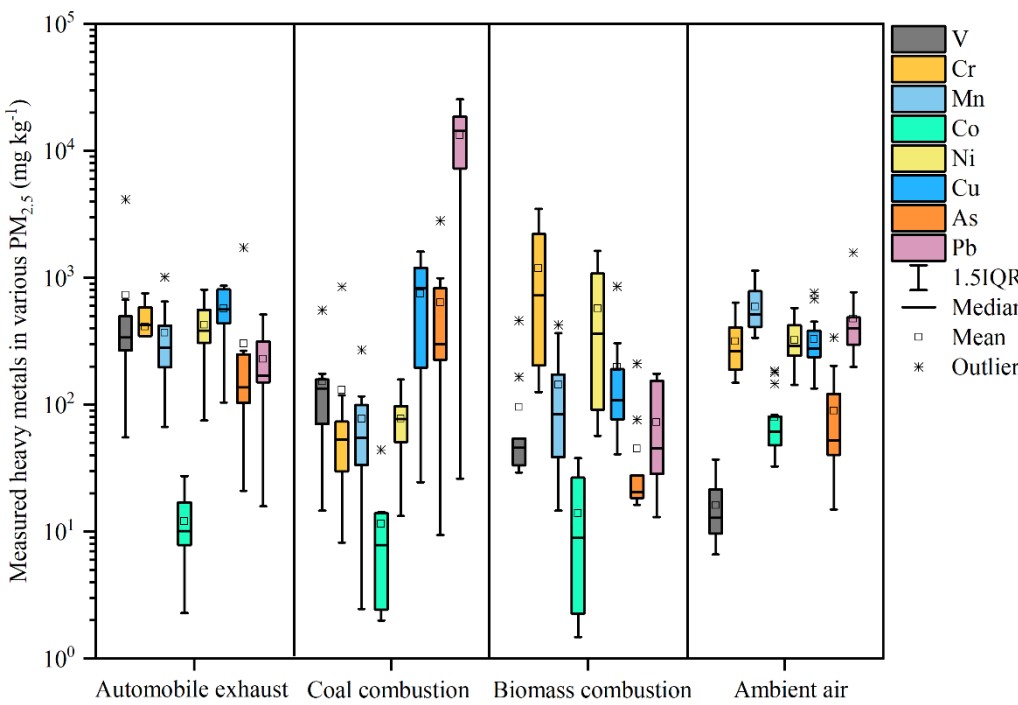


Figure 3: Heavy metal contents (mg kg$^{-1}$) in PM$_{2.5}$ from various specific sources (n=10 for each combustion source and n=16 for urban ambient air).


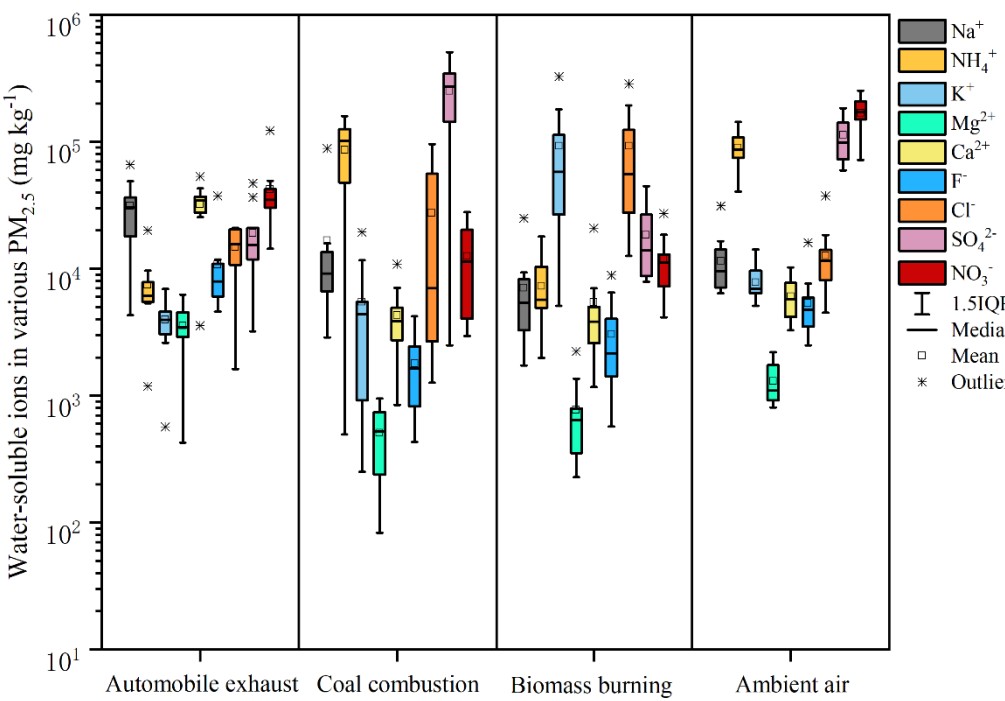


**Figure 4: Water-soluble ion (WSI) contents (mg kg$^{-1}$) in PM$_{2.5}$ from various specific sources (n=10 for each combustion source and n=16 for urban ambient air).**


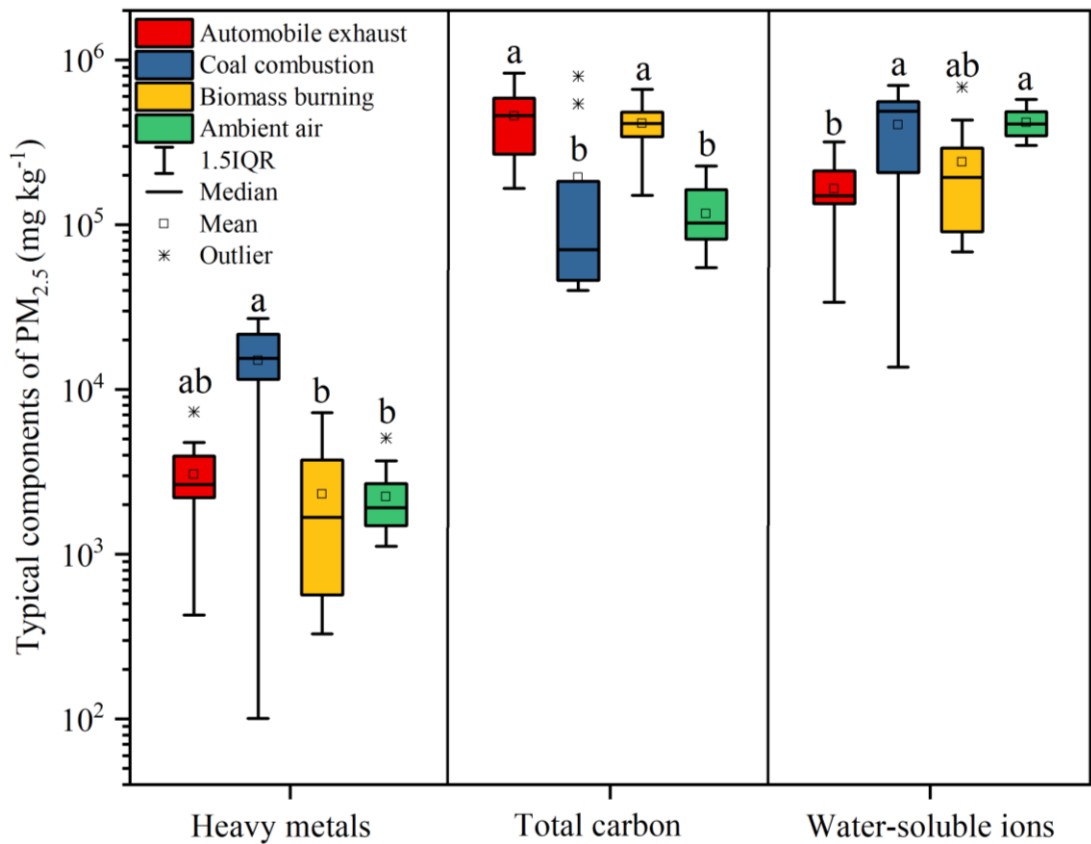


Figure 5: Cumulated typical measured components (mg kg$^{-1}$) in PM$_{2.5}$ from various specific sources (n=10 for each combustion
source and n=16 for urban ambient air). Statistically significant differences between the groups are indicated by different letters
(Kruskal-Wallis test, p < 0.05).

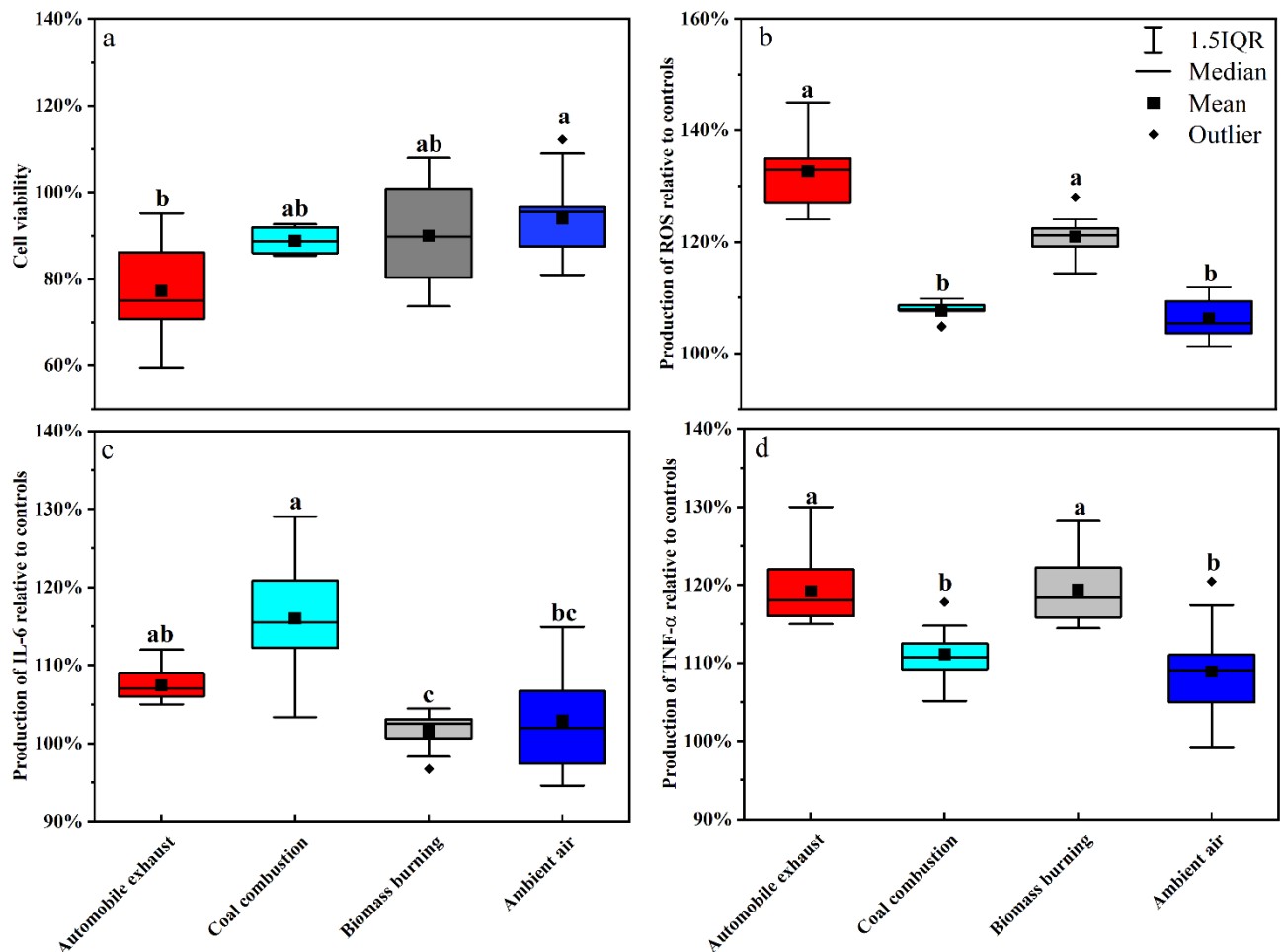


Figure 6: Cell viability, oxidative stress and inflammation levels of human alveolar epithelial cell lines (A549) exposed to PM$_{2.5}$ suspension (80 mg L$^{-1}$) from various specific sources (n=10 for each combustion source and n=16 for urban ambient air). Statistically significant differences between the groups are indicated by different letters (Kruskal-Wallis test, $p < 0.05$).



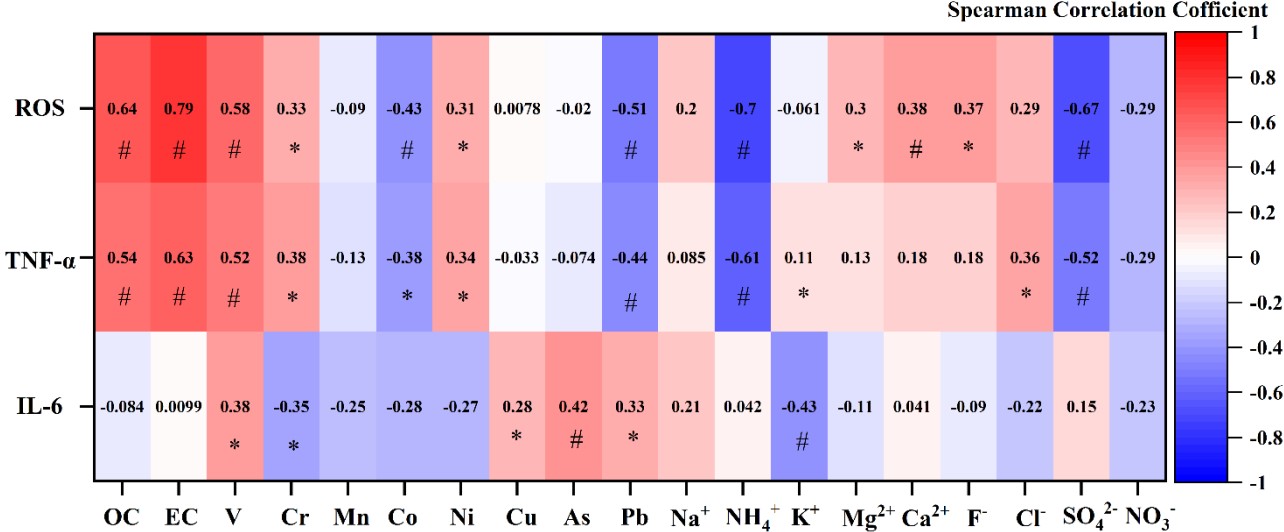


Figure 7: Overall correlations between typical cellular toxicological responses and chemical compositions of PM$_{2.5}$ from various sources (*p < 0.05, #p<0.01; n=46).

