# Peer review of "Source differences in the components and cytotoxicity of PM2.5 from automobile exhaust, coal combustion, and biomass burning contributing to urban aerosol toxicity"

_EGUsphere, 2023_

## Author Comment (AC1)

**Reviewer #1:**

This manuscript has investigated the contribution of three common combustion pollutants to the ambient urban $PM_{2.5}$ health effects. The results are interesting, which showed that particles from different combustion processes at the same concentration exert different toxic effects on the A549 cells. The English language needs to be polished to further improve the quality of this work.

*Reply and revision:*

We appreciate your kindly evaluations very much for our manuscript. This manuscript has been revised thoroughly according to your following advices. To improve the language, we have polished the English carefully overall again. The point-to-point replies and explanations for all revisions are listed below for easy reference.

Other comments are as follows:

lines 24-26: is there any difference between the two words "toxicity" and "toxicogenic" when the authors here used them for different types of samples? Generally, toxicogenic indicated the toxin production activity of bacteria or other organisms.

*Reply and revision:*

Thanks very much for your reminding. Yes, the word "toxicogenic" is inappropriate in current study, and have unified the term by using "toxicity" or "toxic".

line 102: was the Teflon filter also baked in the muffle furnace at 500℃?

*Reply and revision:*

Sorry for the confusion, we describe it more clearly in the revised manuscript. Before being used for sampling, the inorganic quartz filters were incinerated by a muffle furnace at 500 °C for 3 h to remove any possible organic matters, therefore, the parallel $PM_{2.5}$ samples collected by quartz filters could be used for analyzing carbonaceous species. The organic Teflon filters used for collecting parallel $PM_{2.5}$ samples of inorganic analysis were not baked by so high temperature.

Although the air sample information was referred to a literature, it would be better some brief information could be provided here. For example, how about the duration of the air sample?

*Reply and revision:*

Thanks for your reminding. We add some detailed sampling information about the ambient air $PM_{2.5}$ samples in the revised manuscript. As the actual mixture of various source particles in real environment, totally 16 representative ambient air $PM_{2.5}$ samples (each time lasting 23h) covering a year monthly were collected from December 2019 to October 2020 in an urban site surrounded by traffic, residential and commercial quarters of Nanjing city, Yangtze River Delta of eastern China, using a high-volume air sampler (800 L min$^{-1}$) with quartz microfiber filters.

line 136: how about the $PM_{2.5}$ concentrations for the cell stimulation experiments? If 80 mg/L, was the cellular supernatant removed before the addition of $PM_{2.5}$ elution? Cell viability test: has the authors treated the cells with other lower or higher concentrations in addition to the one concentration here (80 mg/L)?

*Reply and revision:*

Yes, the selected concentration of $PM_{2.5}$ suspension is 80 mg L$^{-1}$ based on our pre-experiments covering lower and higher concentrations designed for the dose-response curves. Finally, under this dose, the oxidative stress and inflammation response sensitively, while the cell viability can keep sufficient. The cellular supernatant was removed before the addition of $PM_{2.5}$ elution, so the cells were exposed to the same $PM_{2.5}$ dose.

Correlations between $PM_{2.5}$ components and toxicity: has the authors measured other biological components., e.g., LPS, which is a very strong inflammation inducer and is a common component in the air?

*Reply and revision:*

Much thanks for your nice suggestions. Lipopolysaccharide (LPS) as a common endotoxin in the ambient air is really a strong inflammation inducer, and should be a

significant component from natural sources posing health risks. Because our current study focus on the PM$_{2.5}$ emitted directly from combustion sources, the biological components including LPS in these anthropogenic PM$_{2.5}$ was not measured. But it's sure an important parameter in our future bioaerosols work.

Figure 1: it is not clear the percentage of species in what?

***Reply and revision:***

Thanks for the reminding. We have modified the Figure 1 to indicate the proportion (%) of each component from each source accounting for the corresponding component in urban ambient air PM$_{2.5}$ more clearly.

[Figure]

**Figure 1.** The PMF factor profiles of various components and source percentages of secondary aerosol, automobile exhaust, coal combustion, and biomass burning contributing to the urban ambient air PM$_{2.5}$.

Figure 5: were there any statistically significant difference for each component in different types of samples?

Figure 6: similar to the last question, statistical test?

***Reply and revision:***

Thanks very much for the reminding. We performed a significance analysis based on the Kruskal-Wallis test to modify Figure 5 and 6 in the revised manuscript.

[Figure]

**Figure 5.** Cumulated typical measured components (mg kg$^{-1}$) in PM$_{2.5}$ from various specific sources (n=10 for each combustion source and n=16 for urban ambient air). The letters a and b are significant groups classified by Kruskal–Wallis test, p < 0.05.

[Figure]

**Figure 6.** Cell viability, oxidative stress and inflammation levels of human alveolar epithelial cell lines (A549) exposed to PM$_{2.5}$ suspension (80 mg L$^{-1}$) from various specific sources (n=10 for each combustion source and n=16 for urban ambient air). The letters a, b and c are significant groups classified by Kruskal–Wallis test, p < 0.05.

**Reviewer #2:**

In this work, the authors studied the toxicological responses to $PM_{2.5}$ from different combustion sources. They generated $PM_{2.5}$ from a large variety of sources, including automotive engine exhaust, coal combustion and biomass burning. They concluded that $PM_{2.5}$ in Nanjing is dominated by primary combustion sources, and $PM_{2.5}$ generated from these sources are substantially more toxic than ambient $PM_{2.5}$. In general, the results are interesting and offers insights into PM toxicity. The authors also conducted a broad array of experiments and investigated multiple endpoints relevant to human health. However, the results can be analyzed and discussed in greater depth. The current version of the manuscript does not reflect hypothesis-driven research. The manuscript also contains many grammatical errors and awkward language that is not appropriate for scientific communication, to a point that it hinders reading of the manuscript. The manuscript should not be considered for publication until these substantial issues are fully addressed.

*__Reply and revision:__*

Thanks very much for your critical evaluation on our manuscript, which is an important guidance for us to improve the overall quality of this paper. We checked the whole paper carefully and correct the grammatical errors to make sure the language is accurate, clear and concise. We have revised the manuscript thoroughly according to your following advices on those substantial issues.

Currently, either the world air quality guidelines or the national air quality standards use the mass concentration of $PM_{2.5}$ as the metric for $PM_{2.5}$ pollution evaluation and management, in which all particles are treated as equally toxic, however, it is inconsistent with the scientific facts that particle toxicity are significantly related to their sources and chemical compositions (Shiraiwa et al., 2017; Kelly and Fussell, 2020). Therefore, to identify which component(s) and source(s) of ambient PM are most harmful to health, will be very helpful to optimize air quality guidelines/standards and prioritize targeted PM control strategies to more effectively protect public health.

The hypothesis of this study was that, dominated by the source and component effects, various anthropogenic combustions are very important environmental aerosol sources and contribute different hazardous compositions and thereby unequal toxicity effects to those of the urban ambient $PM_{2.5}$. Therefore, judging the most toxic source as priority emission to be targeted reduced preferentially for precise pollution control, might produce the greatest benefits for public health with improved environmental air quality.

The key results were also analyzed and discussed further in depth, although still limited by the paper length considering large amount of data. Of course, more introduction and discussion about the mechanisms of $PM_{2.5}$ toxicity to lung cells were added in the revised manuscript (also see a detailed answer to the hypotheses question below). The point-to-point replies and explanations for all revisions are listed below for easy reference.

General comments:
This study discusses risks, but is actually investigating hazard when measuring cytotoxicity of PM from specific sources. The exposure levels are all fixed at 80 mg/L. The discussion should better reflect this distinction.

***Reply and revision:***

Thanks very much for your important reminding. Yes, as you suggest, this study investigated the "toxic effects" based on the same mass concentration of $PM_{2.5}$ exposure in body lung fluid system, which should be more precisely than using the word "risks" usually relating to the inhalation exposure concentration of $PM_{2.5}$ in air. So, we unified these descriptions in the revised manuscript. Moreover, for comparing the cytotoxicity, the selected concentration of $PM_{2.5}$ suspension is 80 mg $L^{-1}$ based on our pre-experiments covering lower and higher concentrations designed for the dose-response curves. Finally, under this dose, the oxidative stress and inflammation response sensitively, while the cell viability can keep sufficient for the successful toxicity tests.

What is the justification for using A549 cells? There have been some discussions about the limitations of A549 cells (https://doi.org/10.1016%2Fj.bpj.2009.12.4289) For example, A549 may be more resistant to exposure to external compounds (https://doi.org/10.1158/1078-0432.ccr-08-2822). Often BEAS-2B cells are preferred over A549 cells, even though A549 cells have been used in toxicology studies for many decades. A discussion of the limitations is needed.

***Reply and revision:***

Much thanks for your careful reminding and helpful comments.

Air pollution can harm lung alveoli and epithelial cells. The A549 cell line is derived from human lung cancer and has characteristics similar to alveolar epithelial cells. This cell line has long been used as a suitable epithelial alveolar model to investigate the

interactions between PM and lung epithelial cells (Park et al., 2018; Li et al., 2022b).

Yes, as you suggest, the human normal bronchial epithelial cell BEAS-2B is really preferred over the human lung adenocarcinoma epithelial cell A549, and we honestly selected A549 cell based on our lab's abundant experimental experiences before and also because it has been used popularly in *in vitro* toxicology studies to elucidate the cellular and molecular mechanisms of $PM_s$ involved in lung for many decades. For instance, both cells were used in an aerosol study (Bonetta et al., DNA damage induced by $PM_{0.5}$ samples in A549 and BEAS-2B human cell lines: Results of the MAPEC study. Toxicology Letters, 2017, 280, 1: S208), results of which highlighted the higher sensitivity of BEAS-2B cells respect to A549 also in samples with low level of pollutants, because the PM samples from Italian towns can induce genotoxicity in normal cells while cancer cells might be resistant to their adverse effects. Therefore, although our results are authentic and reasonable in current study, we added some limitations of A549 cells in discussion of the revised manuscript (a new Section 4.4 Limitations), and of course we will choose the generally more sensitive BEAS-2B cells in our subsequent studies.

One of the biggest weaknesses of this study is the lack of central hypotheses. It just reports on a number of endpoints without answering any specific research questions. The central point is that the toxicity measures are the highest for some sources, but there does not seem to be much attempt to answer why. The presentations on the endpoints are not discussed holistically. Why are the cell viability and different inflammatory markers not consistent with each other? What different physiological processes they represent? And how would those be associated or triggered by different trace components? The discussion in Section 4.3 seems to present a singular picture of ROS production, TNF-alpha and IL-6 expression, but the data are more nuanced.

***Reply and revision:***

Thank you very much for the critical comments. More introduction and discussion about the toxicological mechanisms of $PM_{2.5}$ components to lung cells were added in the revised manuscript with significant improvements.

The central hypothesis of this study was that, depending on the source and component effects, various anthropogenic combustions are very significant environmental $PM_{2.5}$ sources and contribute different hazardous compositions and thereby diverse toxicity

effects to those of the urban ambient PM$_{2.5}$. Therefore, by toxicity tests with detailed chemical analyses of these independent source samples, judging the most toxic source as priority emission to be targeted reduced preferentially for precise pollution control, might produce the greatest benefits for public health with improved environmental air quality. So we focused on comparing and quantifying the toxicity of various combustion sourced PM$_{2.5}$ related to the possible mechanisms of toxic components. The toxicity indicated by the endpoints we measured was basic phenomena/discovery and also very valuable knowledge, moreover, we attempted to answer why by their relation with the measured chemical compositions, but it's really not much owing to the focus, and we also can't over-explain only based on statistical relations. We tried more explanation and discussion about the PM$_{2.5}$ toxicological mechanisms in Section 1 (Introduction), Section 2.4 (Cell culture and cellular toxicity tests by in vitro PM$_{2.5}$ exposure), Section 3.3 (Cell viability, oxidative stress and inflammation levels exposed to various mass-normalized PM$_{2.5}$), and Section 4.3 (PM$_{2.5}$ toxicity related to specific sources by pivotal chemical components), with a new Limitation Section 4.4 of the revised manuscript.

The physiological mechanisms of PM-induced cell toxicity in respiratory system have been continuously investigated with some progresses (Kelly and Fussell, 2012, 2020; Shiraiwa et al., 2017; Mack et al., 2020; Li et al., 2022b), such as the metabolic activation, oxidative stress, inflammatory response, and apoptosis, focused on by current study. In brief, after inhalation and deposition onto the epithelium, redox-active materials in PM$_{2.5}$ can induce the release of ROS, which cause oxidative stress (an imbalance between ROS and antioxidants, i.e., disequilibrium of the redox state of a cell) followed by inflammation and cell death. The ROS can mediate subsequent signaling pathways leading to biomolecule damage (e.g., DNA, lipid, and protein) and cellular injury, through mediating inflammatory responses including the release of pro-inflammatory cytokines like IL-6 and TNF-α by epithelial cells (Sabbir Ahmed et al., 2020; Landwehr et al., 2021). For instance, oxidative stress could trigger the induction of pro-inflammatory transcription factors, such as nuclear factor (NF)-κB, via the mitogen-activated protein kinase (MAPK) signaling pathway. Components adsorbed on particle surface, such as redox-active metals (transition metals, Fe, Ni, V, Cr, Cu),

organic compounds (PAHs, quinones), or even carbonaceous core of particles, are responsible for oxidative stress (Cachon et al., 2014; Sabbir Ahmed et al., 2020). The non-redox active metals (Zn, Pb, Al) can also influence the toxic effects of transition metals by exacerbating or lessening the production of free radicals. The EC may not be a directly toxic component of $PM_{2.5}$ but rather operate as a universal carrier of combustion-derived chemicals (semi-volatile organic fractions, transition metals) of varying toxicity. Inorganic soluble sulphates and nitrates are acidic and can interact with and influence the solubility other compositions like metal bioavailability (Kelly and Fussell, 2020).

In this study, multiple biological responses that facilitate identifying the specific PM triggering ROS and inflammatory responses leading to oxidative stress, and cell death were evaluated for source-specific $PM_{2.5}$. Cell viability (metabolic activity) evaluated the mitochondrial dehydrogenase activity of the living cells. Excessive intracellular ROS formation induced by $PM_{2.5}$ is responsible for oxidative stress to the cells. Cytokines IL-6 and TNF-α were determined for the effect of $PM_{2.5}$ on pro-inflammatory response in cells. However, the toxicity of $PM_{2.5}$ may be the result of multiple components acting through different physiological mechanisms, with inconsistent relationships among endpoints (Park et al., 2018). For instance, in BEAS-2B cells, oxidative stress generated by $H_2O_2$ exposure often results in cytotoxicity rather than by stimulating cytokine/chemokine responses, sometimes no correlation between oxidative damage and cytokine/chemokine responses. Moreover, TNF-α gene was not detected in BEAS-2B cells exposed to atmospheric PM collected from Benin, but the gene expression of other inflammatory cytokines (IL-1β, IL-6, and IL-8) were significantly induced, and decreasing cell viability was highly correlated with high secretion of all studied cytokines (Cachon et al., 2014). Therefore, in the present study, it was impossible to analyze all chemicals in $PM_{2.5}$ and determine all related toxicological endpoints, unmeasured chemicals and endpoints might also play roles in the unexplained results.

In toxicity assessments, cell vitality reflects the overall health of cells, encompassing factors such as cell membrane integrity, intracellular metabolic activity, and cell

proliferation capacity. Decreased cellular vitality may be associated with cell damage, toxic effects, or cellular apoptosis. Inflammation markers are employed to assess the extent and nature of inflammatory reactions, including the production of cytokines and inflammatory mediators, as well as the activation status of inflammatory cells. Inflammation is a complex physiological response, typically delineated by the immune and inflammatory reactions of the body to stimuli such as injury or infection. Alterations in inflammation markers can indicate the intensity and nature of the inflammatory response. Consequently, their variations may be incongruous.

Cachon, B. F., Firmin, S., Verdin, A., Ayi-Fanou, L., Billet, S., Cazier, F., Martin, P. J., Aissi, F., Courcot, D., Sanni, A., Shirali, P.: Proinflammatory effects and oxidative stress within human bronchial epithelial cells exposed to atmospheric particulate matter (PM2.5 and PM>2.5) collected from Cotonou, Benin, Environ. Pollut., 185, 340-351, https://doi.org/10.1016/j.envpol.2013.10.026, 2014.

Kelly, F. J., and Fussell, J. C.: Toxicity of airborne particles—established evidence, knowledge gaps and emerging areas of importance, Phil. Trans. R. Soc. A, 378, 20190322, http://dx.doi.org/10.1098/rsta.2019.0322, 2020.

Landwehr, K. R., Hillas, J., Mead-Hunter, R., Brooks, P., King, A., O'Leary, R. A., Kicic, A., Mullins, B. J., Larcombe, A. N.: Fuel feedstock determines biodiesel exhaust toxicity in a human airway epithelial cell exposure model, J. Hazard. Mater., 420, 126637, https://doi.org/10.1016/j.jhazmat.2021.126637, 2021.

Li, T., Yu, Y., Sun, Z., and Duan, J.: A comprehensive understanding of ambient particulate matter and its components on the adverse health effects based from epidemiological and laboratory evidence. Part. Fibre Toxicol., 19, 67, https://doi.org/10.1186/s12989-022-00507-5, 2022b.

Sabbir Ahmed, C.M., Yang, J., Chen, J. Y., Jiang, H., Cullen, C., Karavalakis, G., Lin, Y.-H.: Toxicological responses in human airway epithelial cells (BEAS-2B) exposed to particulate matter emissions from gasoline fuels with varying aromatic and ethanol levels, Sci. Total Environ., 706,135732, https://doi.org/10.1016/j.scitotenv.2019.135732, 2020.

In general, when examining the cell viability, ROS and inflammatory marker production data in the SI, I see significant variability within one source type, greater than between source types. Therefore, conclusions such as this one in the abstract: "The overall cytotoxicity of $PM_{2.5}$ was automobile exhaust > coal combustion > biomass burning, with different toxicity pathways and triggers" is very problematic without examining the statistical significance. I also have some questions about weighting (see comment below) and whether the weighting is reflecting of relative contributions in the atmosphere.

***Reply and revision:***

Thanks for the reminding. We performed a significance analysis based on the Kruskal-

Wallis test to modify the statistical figures and summarisation in the revised manuscript.

Yes, there were really significant variability of toxicological indicators within one PM$_{2.5}$ source type, that is exactly why we selected these 30 representatively specific combustion source samples for mass-normalized investigation independently. These combustion PM$_{2.5}$ samples are different to each other in raw biomass characteristics like compositions, and all the original data of each independent source type were provided by the Figures S8-S19 in the Supplementary Materials. Focusing on the differences among the three combustion source groups, their grouped statistical results for general comparisons were showed in the Figures 2-7 of main text. Considering the multi-endpoints measured and the PM$_{2.5}$ toxicity mechanisms mentioned above, based on the cell viability first, and then ROS followed by inflammatory markers, together with the significantly related toxic chemical composition contents (Park et al., 2018), we put forward a general sequence of overall mass-normalized toxicity for these three combustion sources PM$_{2.5}$ to readers and managers.

As to the weighting contribution, the profiles were averaged with equal weights from each source in Figure 6, which showed the general statistics of the three source groups compared with ambient air, while the independent values of detailed specific types for each source group were showed in Figures S16-S19, respectively. This study investigated the unequal "toxic effects" based on the same mass concentration of PM$_{2.5}$ exposure in body lung fluid system, while the "risks" usually relating to the inhalation exposure concentration of PM$_{2.5}$ in air were not calculated and evaluated much in this paper.

[Figure]

**Figure 6.** Cell viability, oxidative stress and inflammation levels of human alveolar epithelial cell lines (A549) exposed to PM$_{2.5}$ suspension (80 mg L$^{-1}$) from various specific sources (n=10 for each combustion source and n=16 for urban ambient air). The letters a, b and c are significant groups classified by Kruskal–Wallis test, $p < 0.05$.

There has not been any direct evidence that sulfate itself is toxic (or linked with oxidative stress). However, it seems that increase in sulfate is associated with stronger acidity and greater solubility and bioavailability of redox active metals (see for example https://pubs.acs.org/doi/10.1021/acs.est.6b06151). Therefore the conclusion about water soluble ions being a source of toxicity might be too simplistic and should be discussed in more detail.

*__Reply and revision:__*

Thanks for the comment on helping explaining the roles of sulfate in PM$_{2.5}$ toxicity. We involved these discussion and reference in the revised manuscript.

Fang et al., Highly acidic ambient particles, soluble metals, and oxidative potential: a link between sulfate and aerosol toxicity. Environ. Sci. Technol. 2017, 51, 5, 2611–2620.

Specific comments:
Abstract: the abstract should be rewritten with the following considerations. First, the abstract should convey global information, and some of the statements are specific to this study (or geographical area), such as the source apportionment results. These source apportionment results are not applicable to other areas. Furthermore, the abstract does not place the conclusions from the study in the appropriate context. It is unclear how the toxicological results affect PM$_{2.5}$ pollution control policy, which are already targeting PM$_{2.5}$ from major sources, such as coal combustion and vehicle exhaust.

*Reply and revision:*

Thanks very much for your detailed comments. We reorganized the Abstract in the revised manuscript.

Introduction:
I believe these papers are relevant to the investigation of source toxicity and should be discussed in the introduction as part of literature review:
https://doi.org/10.1038/s41598-018-35398-0
https://doi.org/10.1016/j.envpol.2018.09.074

*Reply and revision:*

Much thanks for your reminding. Yes, we read them before but sorry for missing citing. We have added these papers to the introduction and also for result comparisons.

> Park, M., Joo, H. S., Lee, K., Jang, M., Kim, S. D., Kim, I., Borlaza, L. J. S., Lim, H., Shin, H., Chung, K. H., Choi, Y.-H., Park, S. G., Bae, M.-S., Lee, J., Song, H., and Park, K.: Differential toxicities of fine particulate matters from various sources, Scientific Reports, 8, 17007, 10.1038/s41598-018-35398-0, 2018.
>
> Borlaza, L. J. S., Cosep, E. M. R., Kim, S., Lee, K., Joo, H., Park, M., Bate, D., Cayetano, M. G., and Park, K.: Oxidative potential of fine ambient particles in various environments, Environ. Pollut., 243, 1679-1688, https://doi.org/10.1016/j.envpol.2018.09.074, 2018.

The term biomass burning is quite broad and could be better defined. Biomass burning can include biofuel burning (for cookstoves or home heating), agricultural crop burning, wildfires or burning for land use change. It seems that this study is focused on crop burning, based on the type of fuel used. I suggest replace the use of "biomass burning" with "crop burning".

*Reply and revision:*

Thanks for your meticulous comment. Yes, biomass burning can include biofuel burning (both solid, liquid, or gas), agricultural crop burning, and wildfires of plants mainly trees. In this study, we typically burned 8 types of crop (straws of rice, wheat, corn, soybean, peanut, rape, and sesame, corncob), and also 2 types of firewood (branches of peach and pine), so "crop burning" may not cover all biomass investigated. But "plant biomass burning" or "solid biomass burning" might be more accurate and we unified as "plant biomass burning".

Line 33 and others: Many terms (such as particulate matter, consumption) should not be plural

***Reply and revision:***

Corrected overall.

Line 83: I am not sure if "big duty" is the proper term. Maybe "heavy duty"?

***Reply and revision:***

Yes, "heavy duty" should be better and revised.

Line 105: how long were the samples collected for? Are they 24 hour samples?

***Reply and revision:***

Detailed information added. As the actual mixture of various source particles in real environment, totally 16 representative ambient air $PM_{2.5}$ samples (each time lasting 23h) covering a year monthly were collected from December 2019 to October 2020 in an urban site surrounded by traffic, residential and commercial quarters of Nanjing city, Yangtze River Delta of eastern China, using a high-volume air sampler (800 L $min^{-1}$) with quartz microfiber filters.

Line 113: what is the digestion efficiency? What is the digestion method? (I suspect that some of these methodological details are covered in a previous paper, but relevant details should be included in this manuscript.)

***Reply and revision:***

Yes, most methods applied for sample analyses were experienced and published. Detailed information was added. The $PM_{2.5}$ samples were digested by concentrated $HNO_3$-$HClO_4$ acids with a progressive heating program. The recoveries for standard reference material ranged from 90-110 %.

Line 124: ultrasonication has been shown to impact ROS. (https://doi.org/10.1080/02786826.2014.981330) Potential artifacts should be discussed.

*Reply and revision:*

Thanks for the reminding. We cited your recommended paper for reference and made corresponding limitation discussion in the revised manuscript. Because ultrasonication treatment is the most commonly used and even might only efficient method to peel off the particulate matter (PM) from sample filters, the potential impact on ROS of PM can't be completely eliminated at 0 °C and was ignored as a systematic error. Moreover, the experimental indicators of this study did not measure ROS in the PM but cellular ROS.

Miljevic, B., Hedayat, F., Stevanovic, S., Fairfull-Smith, K. E., Bottle, S. E., Ristovski Z. D.: To sonicate or not to sonicate PM filters: reactive oxygen species generation upon ultrasonic irradiation, Aerosol Sci. Tech., 48, 1276-1284, DOI: 10.1080/02786826.2014.981330, 2014.

Lines 143-145: there is an assumption of normality for Pearson correlation test. In many cases the distribution is not necessarily normal, and transformation may be needed (such as a log transform).

*Reply and revision:*

Thank you very much for the nice reminding. According to your comment, we checked the data of normality again and found that Spearman correlation test was more suitable for this study. We also revised the relevant results and discussion in the revised manuscript.

Line 157-159: Secondary organic aerosol often shows a greater toxicity than primary aerosol, and its fraction is still 34% (which is not small). Also, secondary does not mean it is not anthropogenic. Classifying the combustion sources are primary sources (rather than anthropogenic sources) would be more appropriate.

*Reply and revision:*

Yes, you are wright. secondary aerosols can't be ignored, and part of them are also from anthropogenic sources. Our meaning here is exactly that, secondary aerosols (some also from combustions as anthropogenic sources) contribute (34%) to the urban air PM2.5, but the primary sources of combustions as anthropogenic sources contribute more (66%). Sorry for the confusion.

Line 173-174: this sentence is incomprehensible.

*Reply and revision:*

Sorry for the confusion. We have made clear revisions.

Line 180-183: are these results related to differential uptake of metals in different parts of the plant?

*Reply and revision:*

Yes, different plant species and even different plant parts differ significantly in their ability to uptake and accumulate metals from soil (Zhao et al., 2020), that has been explained in the Discussion of Section 4.2.

> Meifang Zhao, Suping Zeng, Shuguang Liu, Zhiqiang Li, Lei Jing, 2020. Metal accumulation by plants growing in China: Capacity, synergy, and moderator effects. Ecological Engineering, 148, 105790, https://doi.org/10.1016/j.ecoleng.2020.105790.

Line 190: why does automotive exhaust contain so much Na and Ca? On road studies would point to these alkali metals coming from resuspended road dust instead, but the experiments here sampled engine exhaust directly.

*Reply and revision:*

Thank for the reminding. Yes, this experiment sampled engine exhaust directly. The contents of $Ca^{2+}$ and $Na^+$ might be related to additives in lubricant oil for anti-wear and anti-corrosion of the engine.

Line 214: why does increased inflammatory injury lead to greater probability of apoptosis? As far as I know, TNF-alpha and IL-6 are not markers of apoptosis, and one cannot distinguish apoptosis from the cell viability results. If the authors are assuming that association based on literature, relevant papers should be cited. Otherwise, none of the results in this study is pointing to apoptosis.

*Reply and revision:*

Thanks for your reminding. We added the following relevant papers cited for reference:

Wang Y, Cao M, Liu A, Di W, Zhao F, Tian Y, Jia J. Changes of inflammatory cytokines and neurotrophins emphasized their roles in hypoxic-ischemic brain damage. Int J Neurosci. 2013,123(3):191-5.

Victor, F. C., and Gottlieb, A. B.: TNF-alpha and apoptosis: implications for the pathogenesis and treatment of psoriasis, J. Drugs Dermatol., 1, 264-275, 2002.

Line 235: why is high NO3- a marker of automotive exhaust? Similar to my previous comment. The Zhang 2022b paper cited in this manuscript only refers to diesel vehicles. Would it be appropriate to apply this finding to all motor vehicles? Without understanding the emissions of these compounds, it would be very difficult to use them as markers confidently.

*__Reply and revision:__*

Thanks for the reminding. Because the concentration of $NO_3^-$ in $PM_{2.5}$ from automotive exhaust in this study was the highest among the three typical combustion sources. We corrected the reference by the following relevant paper:

Hao, Y., Gao, C., Deng, S., Yuan, M., Song, W., Lu, Z., and Qiu, Z.: Chemical characterisation of $PM_{2.5}$ emitted from motor vehicles powered by diesel, gasoline, natural gas and methanol fuel, Sci. Total Environ., 674, 128-139, https://doi.org/10.1016/j.scitotenv.2019.03.410, 2019.

Line 239: road dust is not a natural source

*__Reply and revision:__*

Yes, we corrected as "fugitive soil dust".

Line 244: are the profiles averaged with equal weights from each source, or are they weighted by abundance? For example, there are likely more light duty vehicles than heavy duty vehicles.

*__Reply and revision:__*

Sorry for confusion. Yes, the profiles were averaged with equal weights from each source in Figure 2, which showed the general statistics of the three source groups

compared with ambient air, while the values of detailed specific types for each source group were showed in Figure S4, S5, and S6, respectively. To avoid ambiguity, we have already made clear revisions.

Line 261: heavy metals are not necessarily linked with oxidative stress. More accurately, it should be the "redox-active" metals that are linked with oxidative stress.

*Reply and revision:*

Thanks for the correction. Revised accordingly.

Line 329: the toxicity of PAHs are linked with mutagenicity and not necessarily connected to oxidative stress. Rather, oxidation products of PAHs (such as quinones) are the redox active components. This point here seems quite arbitrary and not necessarily linked with the results shown in this study.

*Reply and revision:*

Yes, you are wright. To avoid ambiguity, we deleted that inappropriate point and re-organized this paragraph.

Line 345: how is this city typical of a megacity in eastern China? In terms of population, or relative source contributions, or climate?

*Reply and revision:*

We revised "typical" as "representative". The three universally typical combustion sources are the most key objects of this study, while the city investigated just provide representative ambient urban air samples. Of course, Nanjing city is a typical megacity in Yangtze River Delta of eastern China, either considering population, socio-economic conditions, air quality, or climate, and the sampling urban site surrounded by traffic, residential and commercial quarters is also very common in China.

Figure 2: how are outliers defined?

In box plots, outliers are usually defined as data points that are outside 1.5 times the interquartile range (IQR) away from the upper and lower quartiles. In this study, the outliers also reflect significant variations within a pollution source type to some extent. Therefore, they were kept in this figure.

Supplementary: Figures S8-S19 would be better presented if the data were grouped by species or endpoints rather than by sources. For example, it would be easier to compare ROS production for different sources if all the ROS data from different sources were right next to each other.

***Reply and revision:***

Thank you very much for the kind suggestion to reclassify the samples by species or endpoints rather than by sources in Figures S8-S19 of the Supplementary Materials. Because all the 30 specific combustion source samples we investigated are selected representatively and are different to each other in raw characteristics like compositions, we provide all the original data of each specific source type showed by the Figures S8-S19 in the Supplementary Materials, and the grouped statistical results for comparisons were showed in the Figures 2-7 of main text. Considering the paper length already with so many figures, and to avoid confusion of too much information in a figure, focusing on the general differences among the three combustion source groups, we finally didn't show the statistical results of sub-groups among each source group in Figures S8-S19, and the specific component was not combined with specific source into a figure.

Of course, we fully understand the information value might be provided by these statistical ways, and we did these analyses for self-use supporting the results and discussion.

---

## Author Response (AR2)

**Reviewers' Comments, Author's Replies, and Revisions in the Text**

(Second Round)

Manuscript Number: egusphere-2023-598

Title: Source differences in the components and cytotoxicity of PM$_{2.5}$ from automobile exhaust, coal combustion, and biomass burning contributing to urban aerosol toxicity

Author(s): Xiao-San Luo, Weijie Huang, Guofeng Shen, Yuting Pang, Mingwei Tang, Weijun Li, Zhen Zhao, Hanhan Li, Yaqian Wei, Longjiao Xie, Tariq Mehmood

MS type: Research article

*Dear editor Dr. Prof. Su and dear reviewers,*

*We are very grateful for your helpful comments again. Based on these kind advices and suggestions, we have made further careful modifications and detailed improvements in depth to the previous manuscript. The changes are shown in the track-changes. The point-to-point replies, explanations, and clarifications for all of the revisions are listed below for easy reference. We have also polished the overall manuscript again.*

*We hope the revised manuscript this time could be published in ACP after these important modifications and significant improvements.*

**Responses to Reviewers' Comments:**

Thanks to the authors for the replies. I just have one more comment. Regarding the statistical analysis in Figure 5 & 6, it is not clear in terms of the letter "a", "b" and "ab". I was wondering about the differences among the different types of samples. For example, was there any significant difference between the ROS production of group automobile exhaust and coal combustion (Figure 6-2)? It would be better if the authors could put a horizontal line above the two compared groups, and label the statistical comparison result with asterisk (statistically significant) or NS (no statistically significant difference). BTW, it is not common to use number "1" to label each panel (Figure 6)

*Reply and revision:*

We appreciate your kindly evaluations again very much for our manuscript. Thanks for the reminding about the marking style of statistical differences. It is possible that the unclear statements of the descriptions about Figure 5 and 6 in our previous manuscript caused the confusion. These figures used letters to indicate the statistical difference between groups, which was significant if they do not contain the same letter, and was not significant if they contain the same letter. The horizontal lines with asterisk or NS above the compared groups is indeed good idea, but if we use this method to indicate the statistical differences between groups in figures of current study, then each set of data would need 5 lines to indicate the significances between different groups, and it might be a bit too crowded in the graphs, so we finally prefer to choose using letters to indicate the difference significances between groups. In addition, according to your nice comments, numbers labeling sub-graph have been replaced in the updated Figure 6.

[Figure]

**Figure 5.** Cumulated typical measured components (mg kg$^{-1}$) in PM$_{2.5}$ from various specific sources (n=10 for each combustion source and n=16 for urban ambient air). Statistically significant differences between the groups are indicated by different letters (Kruskal-Wallis test, p < 0.05).

[Figure]

**Figure 6.** Cell viability, oxidative stress and inflammation levels of human alveolar epithelial cell lines (A549) exposed to PM$_{2.5}$ suspension (80 mg L$^{-1}$) from various specific sources (n=10 for each combustion source and n=16 for urban ambient air). Statistically significant differences between the groups are indicated by different letters (Kruskal-Wallis test, p < 0.05).

The manuscript is an exhausting work considering both the emission tests and toxicity tests on particles emitted from vehicle emission, coal combustion, biomass burning. Also the ambient air was also investigated. The most interesting thing is the toxicity identification of particles emitted from various sources. While the other research points (the toxicity of ambient $PM_{2.5}$ and its source apportionment) are not so impressive as they have been done for a long period. The research contents are a bit of redundant. There are also obvious shortages for the sample collection and PMF modeling. Therefore, I suggested the authors to focus on the toxicity of particles emitted from various sources. It can be accepted after a thorough correction and polishing.

***Reply and revision:***

Thanks very much for your critical evaluation on our manuscript, which is an important guidance for us to improve the overall quality of this paper. It's really an exhausting $PM_{2.5}$ work with abundant both chemical and toxicological results for three major types of combustion sources, and also supported by their final sink - - the ambient air samples. So yes, our aim is focusing on the toxicity of particles emitted from various sources, and many supporting information beneficial to understand the findings are listed in the supplemental materials.

Because last manuscript was the revision after a round of reviewing, we revised and added many contents in accordance with two reviewers' comments during the first round revision. Some comments of the previous round reviewing might be conflicting with this round, but we will integrate them, and have revised the manuscript thoroughly according to your following advices on those substantial issues.

The detailed comments:

1. The authors should not overemphasize the significance of this study, especially in optimizing air quality standards and prioritizing $PM_{2.5}$ control strategies. For optimizing air quality standards and prioritizing $PM_{2.5}$ control strategies, this simple study is quite not enough. Corresponding descriptions should all be corrected, including

the description in Line 15-19, Line 34-36, Line 39-42, Line 48-49, Line 92-93, Line 101, Line 456-457.

*__Reply and revision:__*

Thanks for your comments. This study should be valuable for optimizing air quality standards and prioritizing PM$_{2.5}$ control strategies, but we really still can't overemphasize these significances just by results of our study. We have made the necessary updates to these corresponding statements according to your suggestion, including rewriting and reorganization.

2. For conducting the PM$_{2.5}$ source apportionment, the 16 samples are so limited that I do not believe the authors can obtain a reasonable result or the result can be accepted. 19 chemical components were input into the models, while there are only 16 samples. I suggest the author to use the CMB model, but not the PMF model, as the authors have the source profiles.

*__Reply and revision:__*

Thank you for the reminding and nice suggestion.

The three universally specific combustion sources are the most key objects of this study, while the ambient air is their final sink in the environment, therefore, as the actual mixture of various source particles in real environment, totally 16 typical urban PM$_{2.5}$ samples covering a year monthly were collected to represent the ambient air samples. Although the sample size of ambient PM$_{2.5}$ are limited, as real examples, their chemical and toxicological results could be compared with the source samples, and also imply and explained by the contributions of source samples, indicating the necessity of investigating specific sources samples.

Of course, as you suggested, besides the PMF model applied previously, we further use the CMB model to identify the ambient PM$_{2.5}$ source. We found that the results from the CMB and PMF models are very similar (Figure 1). Therefore, we decided to include a comparison of the results from these two models in the revised manuscript.

The following is a description of the CMB model analysis and results:

Due to the high concentration of sulfate and nitrate in ambient PM$_{2.5}$, and being lack of specific actual source to emit sulfate and nitrate, we added the virtual source profiles of secondary sources in CMB model. The virtual source profiles of secondary sources are represented by the proportion of sulfate, nitrate and ammonium in pure ammonium sulfate and ammonium nitrate. The source profiles of coal combustion, plant biomass burning, automobile exhaust, and secondary sources are shown in the table below.

The parameters of CMB are set as follows: the mass percentage range of PM$_{2.5}$ is 80%-120%, 0< Chi$^2$ ($\chi^2$)<4, and 0.8<R$^2$<1. CMB model identified four major sources of the ambient PM$_{2.5}$, including primary particles of coal combustion, plant biomass burning, automobile exhaust, and secondary aerosols, which account for 20.19%, 8.31%, 25.88%, and 26.07%, respectively. The mass percentage range of PM$_{2.5}$ is 80.45%, Chi$^2$ ($\chi^2$)=1.08, R$^2$=0.82. All parameters show that the CMB model results are well. As the mass percentage range of PM$_{2.5}$ is 80.45%, we further normalized the above four source classes: coal combustion (25.10%); plant biomass burning (10.32%); automobile exhaust (32.17%); secondary aerosols (32.40%).

[Figure]

Inner ring - outer ring:PMF model -CMB model

■ Secondary aerosols ■ Automobile exhaust ■ Coal combustion ■ Biomass burning

**Figure 1.** Source contributions (%) to the urban ambient air PM$_{2.5}$ (PMF model vs CMB model).

**Table S4**. The CMB source profiles of coal combustion, plant biomass burning, automobile exhaust, and secondary sources (g/g).

| Source | Coal combustion | | Biomass burning | | Automobile exhaust | | Sulfate | | Nitrate | |
|---|---|---|---|---|---|---|---|---|---|---|
| | Mean | Standard deviation | Mean | Standard deviation | Mean | Standard deviation | Mean | Standard deviation | Mean | Standard deviation |
| OC | 0.1548 | 0.2046 | 0.2828 | 0.1075 | 0.2159 | 0.0945 | 0.0000 | 0.0000 | 0.0000 | 0.0000 |
| EC | 0.0386 | 0.0609 | 0.1280 | 0.1119 | 0.2420 | 0.1589 | 0.0000 | 0.0000 | 0.0000 | 0.0000 |
| V | 0.0002 | 0.0001 | 0.0001 | 0.0001 | 0.0007 | 0.0011 | 0.0000 | 0.0000 | 0.0000 | 0.0000 |
| Cr | 0.0001 | 0.0002 | 0.0012 | 0.0011 | 0.0004 | 0.0002 | 0.0000 | 0.0000 | 0.0000 | 0.0000 |
| Mn | 0.0001 | 0.0001 | 0.0001 | 0.0001 | 0.0004 | 0.0003 | 0.0000 | 0.0000 | 0.0000 | 0.0000 |
| Co | 0.0000 | 0.0000 | 0.0000 | 0.0000 | 0.0000 | 0.0000 | 0.0000 | 0.0000 | 0.0000 | 0.0000 |
| Ni | 0.0001 | 0.0000 | 0.0006 | 0.0005 | 0.0004 | 0.0002 | 0.0000 | 0.0000 | 0.0000 | 0.0000 |
| Cu | 0.0007 | 0.0006 | 0.0002 | 0.0002 | 0.0006 | 0.0002 | 0.0000 | 0.0000 | 0.0000 | 0.0000 |
| As | 0.0006 | 0.0008 | 0.0000 | 0.0001 | 0.0003 | 0.0005 | 0.0000 | 0.0000 | 0.0000 | 0.0000 |
| Pb | 0.0132 | 0.0083 | 0.0001 | 0.0001 | 0.0002 | 0.0001 | 0.0000 | 0.0000 | 0.0000 | 0.0000 |
| $Na^+$ | 0.0166 | 0.0243 | 0.0071 | 0.0064 | 0.0312 | 0.0166 | 0.0000 | 0.0000 | 0.0000 | 0.0000 |
| $NH_4^+$ | 0.0860 | 0.0546 | 0.0073 | 0.0046 | 0.0074 | 0.0047 | 0.2730 | 0.0273 | 0.2250 | 0.0225 |
| $K^+$ | 0.0055 | 0.0056 | 0.0929 | 0.0930 | 0.0039 | 0.0016 | 0.0000 | 0.0000 | 0.0000 | 0.0000 |
| $Mg^{2+}$ | 0.0005 | 0.0003 | 0.0008 | 0.0006 | 0.0036 | 0.0015 | 0.0000 | 0.0000 | 0.0000 | 0.0000 |
| $Ca^{2+}$ | 0.0043 | 0.0027 | 0.0055 | 0.0054 | 0.0322 | 0.0123 | 0.0000 | 0.0000 | 0.0000 | 0.0000 |
| $F^-$ | 0.0018 | 0.0012 | 0.0031 | 0.0025 | 0.0108 | 0.0091 | 0.0000 | 0.0000 | 0.0000 | 0.0000 |
| $Cl^-$ | 0.0273 | 0.0347 | 0.0928 | 0.0841 | 0.0148 | 0.0059 | 0.0000 | 0.0000 | 0.0000 | 0.0000 |
| $SO_4^{2-}$ | 0.2504 | 0.1529 | 0.0185 | 0.0111 | 0.0191 | 0.0125 | 0.7270 | 0.0727 | 0.0000 | 0.0000 |
| $NO_3^-$ | 0.0125 | 0.0081 | 0.0119 | 0.0065 | 0.0421 | 0.0283 | 0.0000 | 0.0000 | 0.7750 | 0.0775 |

3. In the abstract, the authors indicated that the contributions of $PM_{2.5}$ from combustion sources to the health risks are not unclear. I think it is not suitable, as former studies have conducted works on evaluate the health risks of $PM_{2.5}$ from various sources, though they combined the source apportionment results and carcinogenic and non-carcer risks calculated by the USEPA equations. I believe the toxicity of chemical components in $PM_{2.5}$ from some sources has also been conducted, but not so comprehensive like this study. Corresponding studies should be thoroughly summarized.

*__Reply and revision:__*

Sorry for the inaccurate statements leading misunderstanding. Indeed, former studies

have conducted works on evaluate the health risks of $PM_{2.5}$ from some sources, mainly calculated by the human health risk assessment equations, but lack of considering the important different toxic roles of various chemical components in $PM_{2.5}$ related to pollution sources, which is one major trigger and objective of our study. According to your suggestion, we reorganized the corresponding descriptions. Corresponding literature were summarized and added in Introduction.

4. Line 34-36, the conclusion is not so meaningful. Without this study, the emission control measures including strengthening the emission standards, coal to gas and coal to electricity, cutting off the crop straw burning have been proposed. The authors should give more accurate conclusions through the key findings of this study, but not repeat formers.

*Reply and revision:*

Thanks for your comments. We have rewritten these sentences showing key findings of this study based on the specific source toxicity comparisons.

5. I do not agree with the authors that the world air quality guidelines treated the $PM_{2.5}$ as equally toxic. The establishment of air quality standards relied on the former toxicity and epidemiology researches, and they are not established by imaging. The WHO and abundant scientists know the toxicity of various chemical components. To establish a standard, lots of experiments should be done, and the performability should also be considered. I suggest the authors to read some papers to learn how to establish an air quality standard for air pollutants.

*Reply and revision:*

Thanks very much for your kindly reminding. We corrected and revised all corresponding statements to be more accurate by avoiding misunderstanding.

6. Line 51-56, we all know these knowledges and the sources of aerosol are not the key research points of this study. One sentence is enough. The toxicity of aerosols from various sources should be more summarized.

*Reply and revision:*

We have polished and condensed these contents according to your comments in the revision. Owing to the background information required for some readers, necessary sentences were kept to make the logic of this study clear.

7. Line 73-74, I don't agree with this description. The heavy metals and PAHs in $PM_{2.5}$ have been listed as the toxic components by the main countries, which means that their toxicity have been tested. Additionally, there are so many studies on the health risk assessment of heavy metals and PAHs (by USEPA methods) and the source apportionment of health risks published, the reviewers even do not want to read such kinds of papers.

*Reply and revision:*

We corrected and revised the inappropriate statements according to your comments. Yes, we know that the pure chemicals of heavy metals and PAHs are well known toxic pollutants with corresponding tested toxicity and have been widely monitored in environmental managements including air pollution, and there are really abundant routine papers reporting their health risks just assessed by the USEPA calculation methods, but the exposure to human lung cells of these mixed chemicals bound in $PM_{2.5}$ inducing toxicity by inhalation are still not clear. Moreover, air particles are so complicated mixture of many chemicals, such as the sulfate and nitrate salts, not only the general EPA listed pollutants available in various environmental media, their toxicity effects and mechanisms is really a significant question.

8. Line 79-80, how the source profiles be applied in elucidate the aerosol pollution and control strategies? The source profiles of different emission inventories indicated what? The emission inventory was established by activity data and emission factors. Source profiles do not belong to emission inventories.

*Reply and revision:*

Thank you for the reminding. We revised these statements. Due to variations in particle composition, sources, and toxicity in different urban environments, it is necessary to

establish aerosol source emission inventories for different regions to elucidate local aerosol pollution characteristics and facilitate control strategies.

9. Line 81-89, the logic is confusing. For example, the author indicated that straw burning contributed to air pollution in many regions, then the author indicated that the aerosol from biomass burning in Amazon had an ability to induce ROS. Do the author want to say the aerosol from straw burning could induce ROS, or the aerosol from biomass burning in Amazon contributed to air pollution in many regions? Meanwhile, the many regions including where? The authors should give more accurate descriptions. Such kinds of descriptions are ubiquitous in this study.

*Reply and revision:*

Thanks for your reminding. We checked and revised the statements overall by more accurate descriptions to avoid confusions.

10. Line 85-86, the authors listed the sulfate is mainly from coal combustion, but not mention its toxicity. This example did not contain the similar meanings with other sentences.

*Reply and revision:*

Because a reviewer of last round has previously mentioned that there was not direct evidence suggesting sulfate itself is toxic or associated with oxidative stress, toxicity is not mentioned in this context. Our aim was twofold: firstly, to illustrate the detrimental effects of anthropogenic sources such as vehicle exhaust, coal combustion, and biomass burning on human health through these examples; and secondly, to emphasize that researchers often focused exclusively on one specific source or component, rarely comparing multiple sources with detailed components together as done in this work.

11. Line 94, the source samples are not abundant in the field of source profiles and emission factors studies.

*Reply and revision:*

Thanks for your reminding. We have replaced the word "abundant" with the specific

number of samples.

12. Line 106, line 109-110, Line 120-123, Line 145, Line 165-167 are redundant. The authors should describe the sampling and analysis methods directly in this section, do not list such inessential information.

*Reply and revision:*

Because some detailed sampling information in this part were requested by the reviewers of last round reviewing, we polished and simplified these contents according to your comments.

13. Line 118, characteristic to physical-chemcial

*Reply and revision:*

Revised.

14. Line 127-130, there are four channels, the 160 L min-1 indicated each channel or the four channels together? What is the dilution ratio for the burning test? For each type of fuels, how many times were repeated? How many fuels were burned in each test? The detailed sampling information should be given. Same information for other combustion tests should also be given.

*Reply and revision:*

We clarified these statements. Some information has been showed in Tables of the Supplemental Materials, and we included more detailed information in Table S1-S3 of the revised manuscript.

**Table S1.** Characteristics and collection process of the investigated typical vehicles.

| No. | Abbreviations | Vehicle types | Manufacture year | Emission standards | Fuel type | Collection time (min) | Weight (kg) |
|---|---|---|---|---|---|---|---|
| #1 | SDGCs-1 | Small duty gasoline coach | 2015 | CN.V | CN.92 # | 120 | 1970 |
| #2 | SDGCs-2 | Small duty gasoline coach | 2019 | CN.VI | CN.92 # | 120 | 2110 |
| #3 | SDDCs | Small duty diesel coach | lost | CN.IV | CN.5# | 20 | 1790 |
| #4 | MDDCs | Middle duty diesel coach | 2009 | CN.IV | CN.5# | 20 | 3600 |

| #5 | HDDCs | Heavy duty diesel coach | 2015 | CN.V | CN.5# | 20 | 15800 |
| #6 | LDDVs-1 | Light duty diesel van | 2009 | CN.III | CN.5# | 20 | 3970 |
| #7 | LDDVs-2 | Light duty diesel van | 2015 | CN.IV | CN.5# | 20 | 4500 |
| #8 | MDDVs | Middle duty diesel van | 2014 | CN.IV | CN.5# | 20 | 7320 |
| #9 | HDDVs-1 | Heavy duty diesel van | 2015 | CN.IV | CN.5# | 20 | 29080 |
| #10 | HDDVs-2 | Heavy duty diesel van | 2019 | CN.V | CN.5# | 20 | 40000 |

**Table S2.** Characteristic analysis and collection process of typical coal samples.

| Coal types | $M_{ad}$ (%) | $A_{ad}$ (%) | $V_{ad}$ (%) | $FC_{ad}$ (%) | Fuel consumption (g) | Burning duration (min) | Origin |
| --- | --- | --- | --- | --- | --- | --- | --- |
| HC-1 | 1.87 | 46.2 | 9.87 | 42.1 | 1169 | 158 | Nanjing city |
| HC-2 | 2.15 | 49.3 | 9.63 | 38.9 | 1138 | 144 | Nanjing city |
| AC-1 | 1.26 | 10.2 | 10.6 | 78 | 739 | 222 | Ningxia province |
| AC-2 | 1.19 | 12.5 | 10.8 | 75.5 | 1024 | 180 | Anhui province |
| AC-3 | 1.76 | 6.78 | 8.99 | 82.5 | 537 | 170 | Shanxi province |
| BC-1 | 5.23 | 1.84 | 41.5 | 51.5 | 8117 | 102 | Inner Mongolia province |
| BC-2 | 7.06 | 5.07 | 29.8 | 58 | 669 | 85 | Henan province |
| IC-1 | 0.43 | 13 | 1.63 | 85 | 559 | 115 | Nanjing Iron & Steel Co. |
| IC-2 | 1.74 | 11.1 | 30.3 | 56.9 | 601 | 90 | China Resources Jiangsu Nanre Power Generation Co. |
| IC-3 | 4.37 | 8.17 | 30.9 | 56.5 | 652 | 95 | Huaneng Nanjing Jinling Power Generation Co. |

Note: $M_{ad}$ is the moisture mass fraction of the sample on an air-dried basis; $A_{ad}$ is the ash mass fraction of the sample on an air-dried basis; $Vd_{ad}$ is volatile matter mass fraction of sample on dry air-dried basis; $FC_{ad}$ is fixed carbon fraction of the sample on an air-dried basis; $FC_{ad} = 1 - M_{ad} - A_{ad} - Vd_{ad}$.

**Table S3.** Characteristic analysis and collection process of typical plant biomass fuel samples.

| Biomass types | $M_{ad}$ (%) | $A_{ad}$ (%) | $V_{ad}$ (%) | $FC_{ad}$ (%) | Fuel consumption (g) | Burning duration |
| --- | --- | --- | --- | --- | --- | --- |
| Rice straw | 10.8 | 14.6 | 59.8 | 14.9 | 83 | 4′24″ |
| Wheat straw | 12.1 | 5.65 | 65.5 | 16.8 | 328 | 9′14″ |
| Corn straw | 11.6 | 4.22 | 66.1 | 18.1 | 108 | 4′39″ |
| Soybean straw | 11 | 4.62 | 68.4 | 16 | 360 | 11′24″ |
| Peanut straw | 15 | 10.8 | 61.4 | 12.8 | 49 | 1′20″ |
| Rape straw | 11.1 | 2.95 | 68.8 | 17.1 | 39 | 1′05″ |
| Sesame straw | 13.1 | 7.64 | 63.7 | 15.5 | 154 | 2′42″ |
| Corncob | 9.21 | 0.66 | 73.5 | 16.7 | 131 | 11′35″ |
| Pine branches | 13.4 | 0.33 | 66.6 | 19.7 | 148 | 12′20″ |
| Peach branches | 9.94 | 0.65 | 73.4 | 16 | 244 | 16′45″ |

Note: $M_{ad}$ is the moisture mass fraction of the sample on an air-dried basis; $A_{ad}$ is the ash mass fraction of the sample on an air-dried basis; $Vd_{ad}$ is volatile matter mass fraction of sample on dry air-dried basis; $FC_{ad}$ is fixed carbon fraction of the sample on an air-dried basis; $FC_{ad} = 1 - M_{ad} - A_{ad} - Vd_{ad}$.

15. Line 135-136, what is the diameter of the filter? How many filters were cut for

carbon, ion and toxicity tests, respectively?

*Reply and revision:*

The diameter of all filters is 47 mm. Carbon, ion, heavy metal, and toxicity testing each require one parallel sample filter, that uses a total of four filters across the four parallel channels.

16. Line 137, why the author said that 16 samples were representative? Representative of what?

*Reply and revision:*

The three universally typical combustion sources are the most key objects of this study, while the 16 ambient $PM_{2.5}$ samples investigated just provide representative real urban air samples in environment. In last round revision, we revised "typical" as "representative". As the actual mixture of various source particles in real environment, totally 16 example ambient air $PM_{2.5}$ samples (each time lasting a day) spanning all months and different seasons of a year were collected in an urban site surrounded by traffic, residential and commercial quarters of a typical megacity, which site is also very common in China and even globally. Although the ambient $PM_{2.5}$ sample number is limited, these environmental samples can still provide insights into the temporal variations in urban air quality related to the investigated sources and compositions. Undoubtedly, a larger sample size of ambient $PM_{2.5}$ would be much better for the study focusing on the ambient aerosols.

17. Line 141-143, the sentence is not essential.

*Reply and revision:*

Revised.

18. Line 148, some elements indicated which elements?

*Reply and revision:*

Revised with details.

19. Line 157, 1/2 or 1/4 or other fraction of filter was cut into pieces?

*Reply and revision:*

Details were added. Due to the small diameter (47 mm) of filters, each chemical analysis and toxicity testing require one whole parallel sample filter.

20. Line 163, medium (DMEM) medium repeat

*Reply and revision:*

Revised.

21. Line 193, why these species were selected for PMF modeling?

*Reply and revision:*

These components represent all the measured chemical data we obtained, and are also commonly used as source tracers in source apportionment studies.

22. Line 199, how the author got the daily $PM_{2.5}$ concentrations? You should give clearly information. Did the author just used the 16 filter samples to calculate its mass concentration? Exceeded the healthy guidelines obviously indicated exceeded how many times?

*Reply and revision:*

The word "daily" in this paper is not "every day" but "a day". As a routine standard method, we calculated the daily ambient $PM_{2.5}$ concentration through gravimetric measurement of each filter and sampled air volume lasting 23h for a day. Moreover, specific comparison results with healthy guidelines in Fig S3 were showed in the revision.

23. Plant biomass burning can be changed into domestic biofuel burning

*Reply and revision:*

Thanks for your nice suggestion. Yes, biomass burning can include biofuel burning (both solid, liquid, or gas), agricultural crop burning (both open and domestic), and wildfires of plants mainly trees. In this study, we typically burned 8 types of crop

(straws of rice, wheat, corn, soybean, peanut, rape, and sesame, corncob), and also 2 types of firewood (branches of peach and pine), so "domestic biofuel burning" may not match all biomass investigated, and "plant biomass burning" or "solid biomass burning" might be more accurate. Because we focus on domestic solid biofuel burning in this study, we replaced "plant biomass burning" with "plant biomass (domestic biofuel) burning" or " domestic plant biomass burning" for corresponding statements.

24. Line 207-209 can be deleted directly. The authors should give the results directly. Please do not give repeat or meaningless information. This question should be corrected for the whole manuscript.

*Reply and revision:*

Thanks for your kind reminding. We simplified the manuscript overall to keep streamlined.

25. Line 213, why it indicated that the OC in ambient $PM_{2.5}$ was lower. It may also indicate that the OC in the ambient air may be aged or cleaned. The authors should read more papers on the atmospheric chemistry of OC.

*Reply and revision:*

Thanks for the comments. We added more discussions according to your suggestions.

26. Line255-257, Line 291, Line 292, Line 293, ….much higher, higher…. The authors should give quantitative description. Such problem should be corrected for the whole paper.

*Reply and revision:*

Thanks for your kind reminding. We checked the whole paper again and used quantitative descriptions in the revised manuscript.

27. Line 264-265, of course the anthropogenic combustion sources should be controlled. without this study, we all know this.

*Reply and revision:*

According to your comment, we simplified general knowledge and summarized new findings to make the revised manuscript more concise and succinct.

28. Line 276, they are not new markers.

***Reply and revision:***

We revised the description.

29. Line 286-288, the sentence is meaningless.

***Reply and revision:***

This sentence was revised, which was a remark explaining the limitations of PMF source models applied in current study owing to the absence of natural sources.

30. Line 296-297, the volatile fraction is composed of organic matter. Of course, it is composed of organic matter. The description is right, but so boring. I suggest the author read more papers to give the formation mechanisms of OC during combustion sources and to support the higher TC contents from coal burning.

***Reply and revision:***

Thanks for your suggestion. We simplified the general knowledge. For the formation mechanisms of OC, we also added more discussion and reference on coal combustion in the revised manuscript. Below are the additional references added:

He K, Shen Z, Zhang B, et al. Emission profiles of volatile organic compounds from various geological maturity coal and its clean coal briquetting in China. Atmospheric Research, 2022, 274: 106200.

Zhou W, Jiang J, Duan L, et al. Evolution of submicrometer organic aerosols during a complete residential coal combustion process. Environmental Science & Technology, 2016, 50(14): 7861-7869.

De la Puente G, Iglesias M J, Fuente E, et al. Changes in the structure of coals of different rank due to oxidation effects on pyrolysis behaviour. Journal of Analytical and Applied Pyrolysis, 1998, 47(1): 33-42.

31. Line 302, there were only 16 samples, one for one month. Of course, they varied seasonally. Another thing is not reasonable is that one sample can not be represent for one month and three samples can not be represent for one season. I can not agree with this. Other similar description and conclusion could also not be accepted.

*Reply and revision:*

Thank you for your suggestion, we have modified the statements about ambient samples in the revised manuscript by providing a more comprehensive perspective to avoid arbitrary expressions. Totally 16 ambient $PM_{2.5}$ samples were indeed limited in representing a year, but in current exhausting study focusing on the various combustion source samples, we can only choose representative real air samples covering each moth and season to support the main source research meaningfully as the mixture sink of their contributions. In fact, for ambient air $PM_{2.5}$, we have collected massive samples frequently in four different sites of this megacity lasting 8 years, but the corresponding research for these abundant ambient samples with spatial-temporal characteristics would be another exhausting valuable work in future.

32. Line 306-307, the author drew the conclusion subjectively. My question is that how to control EC from diesel vehicles? In line 330-332 and in line 347-348, how to control these elements and ions in the particles emitted from these sources selectively. Can you tell us the method?

*Reply and revision:*

Thanks for your critical comments, we made revision in accordance. The exact effective methods to control these specific key toxic components from the emissions of various combustion sources indeed a challenge, but need to be explored. The chemical findings of our toxicological research point a specific direction for better air pollution control, that should be helpful for the environmental technology with potential methods, targeting source materials, combustion processes, or final emissions. Both the basic findings of this study and the corresponding technological solutions not investigated in this manuscript would be valuable for the clean air and public health. Moreover, environmental management policies might also be beneficial to such aims, such as the

choice of fuel types.

33. Line 397, are you sure, the metals and ions can be controlled by strengthening the emission standards? For biomass burning, the particles may all hold high concentrations of OC, Cl⁻ and K⁺, how to control them?

*Reply and revision:*

Thanks for your reminding about the accurate descriptions. We made corresponding revisions. The measures mentioned have an overall impact on source emissions. Besides the environmental technological methods of controlling toxic components targeting source materials, combustion processes, and final emissions, the environmental management policies are also beneficial to this aim, such as the choice of fuel types, especially for the management of domestic biomass fuel burning. For examples, potential measures include promoting new green energy vehicles and low-ash clean coals, depressing the diesel exhaust and rural crop straw burning emissions.

34. Line 438, I think this sentence say nothing, and we all know these knowledges.

*Reply and revision:*

According to your comment, we simplified the general knowledges and summarized new findings.

35. Line 442, the $PM_{2.5}$ can not reflect the air situation of eastern China. The site is just a site in a megacity. There are totally 16 samples and the source apportionment results are quite inconvincible.

*Reply and revision:*

We have updated the relevant descriptions. Yes, we know the limitations of these 16 ambient samples and just use them for comparisons with specific source samples and representation of mixed multi-source samples in real environment. The source apportionment results were cross-validated by models PMF and CMB, supporting the main objectives of this study focusing on various combustion source samples.

36. Line 456-457, the authors just compared the compositions and toxicity of PM$_{2.5}$ from some types of sources, and the manuscript can not provide supports for establishing economical composition-source-based strategies for aerosol pollution control.

***Reply and revision:***

Thanks for your kind reminding. We restricted the description accordingly.

37. References: the format should be corrected, such as the subscript and low level mistakes

***Reply and revision:***

We have checked and updated all references accordingly.

38. Figure S1, what is the dwell room? Size selector should be corrected into PM$_{2.5}$ cut inlets

***Reply and revision:***

We employ the term "residence chamber" as a substitute for the phrase "dwell room". The residence chamber constitutes a crucial element of the dilution tunnel sampler, serving the purpose of allowing the diluted and mixed exhaust to dwell for a specific duration. This facilitates the exhaust cooling and mixture of condensable particles at proper concentrations.

[Figure]

**Figure S1.** Schematic of a dilution 4-channel sampler used for collecting PM$_{2.5}$ directly from various combustion source emissions.

39. Figure S3, I suggest the authors use the air quality monitoring dataset from local environmental monitoring station for the seasonal comparison. The dot-line figure should be corrected into column figure as the data is not continuous.

*Reply and revision:*

Revised. The dot-line figure has been replaced with column figure. Considering the main objective of current study, we analyze the monitored data by the self-collected samples with constrained statements.

[Figure]

Figure S3. Example of daily urban air PM$_{2.5}$ concentrations (μg m$^{-3}$) monitored in Nanjing city, eastern China.

40. Figure S4-S10, Figure S12-S15, did the authors do parallel samples? Can the standard errors be given?

***Reply and revision:***

Thanks for the reminding. Owing to the huge cost of this exhausting work including several chemical and toxicological parameters for abundant samples, we did not analyze parallel samples for every samples. For quality control measures, reference materials were adopted and analytical experiments were performed only after recovery was achieved.

41. Figure S11, it is not monthly PM$_{2.5}$, and it is just PM$_{2.5}$ sample for the selected days.

***Reply and revision:***

We revised the description about samples from each month according to your comment.